# Effectiveness of dual active ingredient insecticide-treated nets in preventing malaria: A systematic review and meta-analysis

Timothy Hugh Barker[1]*, Jennifer C. Stone[1]*, Sabira Hasanoff[1], Carrie Price[2], Alinune Kabaghe[3], Zachary Munn[1]

**1** JBI, Faculty of Health and Medical Sciences, University of Adelaide, Adelaide, South Australia, Australia, **2** Albert S. Cook Library, Towson University, Towson, Maryland, United States of America, **3** Training and Research Unit of Excellence, Blantyre, Malawi

\* timothy.barker@adelaide.edu.au (THB); jennifer.stone@adelaide.edu.au (JCS)

**Data Availability Statement:** All relevant data are within the paper and its Supporting Information files.

## Abstract

Malaria vectors have demonstrated resistance to pyrethroid-based insecticides used in insecticide-treated nets, diminishing their effectiveness. This systematic review and meta-analysis investigated two forms of dual active-ingredient (DAI) insecticide-treated nets (ITN (s)) for malaria prevention. A comprehensive search was conducted on July 6th 2022. The databases searched included PubMed, Embase, CINAHL, amongst others. Trials were eligible if they were conducted in a region with ongoing malaria transmission. The first DAI ITN investigated were those that combined a pyrethroid with a non-pyrethroid insecticides. The second DAI ITN investigated were that combined a pyrethroid with an insect growth regulator. These interventions were compared against either a pyrethroid-only ITN, or ITNs treated with pyrethroid and piperonyl-butoxide. Assessment of risk of bias was conducted in duplicate using the Cochrane risk of bias 2 tool for cluster-randomised trials. Summary data was extracted using a custom data-extraction instrument. This was conducted by authors THB, JCS and SH. Malaria case incidence was the primary outcome and has been meta-analysed, adverse events were narratively synthesised. The review protocol is registered on PROSPERO (CRD42022333044). From 9494 records, 48 reports were screened and 13 reports for three studies were included. These studies contained data from 186 clusters and all reported a low risk of bias. Compared to pyrethroid-only ITNs, clusters that received pyrethroid-non-pyrethroid DAI ITNs were associated with 305 fewer cases per 1000-person years (from 380 fewer cases to 216 fewer cases) (IRR = 0.55, 95%CI: 0.44–0.68). However, this trend was not observed in clusters that received pyrethroid-insect growth regulator ITNs compared to pyrethroid-only ITNs (from 280 fewer cases to 135 more) (IRR = 0.90, 95%CI: 0.73–1.13). Pyrethroid-non-pyrethroid DAI ITNs demonstrated consistent reductions in malaria case incidence and other outcomes across multiple comparisons. Pyrethroid-non-pyrethroid DAI ITNs may present a novel intervention for the control of malaria.

**Funding:** This work was funded by the World Health Organisation, APW202903809 The funder of the study had a role in the development of the protocol, the wording and development of the review questions, the interpretation of the final results and the development of this manuscript.

**Competing interests:** The authors have declared that no competing interests exist.

# Introduction

Malaria is an infectious, parasitic disease transmitted through the bite of infected female *Anopheles* mosquitoes [1]. Malaria is caused by the *Plasmodium* parasite, with *P. falciparum* and *P. vivax* species being the most virulent and widespread for human hosts [1]. Malaria presents a significant burden to global public health, with an estimated 247 million malaria cases in 2021 [2] Substantial progress has been made since 2000 in reducing global malaria cases from 80 cases per 1000 persons at risk to 57 per 1000 persons at risk in 2019. However, there was recently an increase in this metric to 59 per 1000 persons at risk observed in 2020 [2]. The most successful malaria prevention strategies have often included the distribution of insecticide-treated nets (ITN) distribution of ITNs is estimated to have contributed an estimated 68% to the reduction of the malaria burden [2].

The WHO recommends that ITNs treated with a pyrethroid-based insecticide be used for large-scale deployment [3]. These ITNs are prequalified by WHO and are treated with pyrethroid at the time of manufacture and have demonstrated public health value whilst meeting safety standards. However, recent findings have demonstrated that both *Anopheles gambiae* (*s. s.*) and *An. funestus* (*s.s.*), the most prevalent malaria vectors, have developed widespread resistance to these pyrethroid insecticides [4, 5]. This may compromise the long-term effectiveness of these ITNs [1]. In response to the spread of pyrethroid resistance, the WHO has stated that new types of ITNs should be developed to combat insecticide-resistant vectors [3]. WHO has identified two additional classes of ITNs, those designed to kill host-seeking insecticide-resistant mosquitoes and those designed to sterilize and/or reduce their fecundity.

The former of these additional ITN classes, includes ITNs designed to kill resistant mosquitoes and consist of combinations of pyrethroid insecticides and other active ingredients. Belonging to this class includes ITNs treated with both a pyrethroid and piperonyl butoxide (PBO) [1]. PBO is a synergist that acts to inhibit the metabolic enzymes of the mosquito that work to detoxify (and therefore reduce effectiveness of) insecticides. The benefits to public health of these pyrethroid-PBO ITNs have been demonstrated, resulting in the WHO conditionally recommending that these nets be used, particularly in areas where pyrethroid-resistant mosquitoes are present [1]. This class also provisionally includes ITNs that combine pyrethroids with other non-pyrethroid active ingredients (henceforth referred to as dual active ingredient nets, DAI). Studies on one DAI ITN, that combines alpha-cypermethrin (pyrethroid) and the pyrrole chlorfenapyr have recently demonstrated both entomological and epidemiological benefit [6, 7]. Finally, the third class of ITNs include those that have been designed to sterilize and/or reduce the fecundity of host-seeking insecticide-resistant mosquitoes. This class provisionally includes DAI ITNs treated with a pyrethroid insecticide and an insect growth regulator such as pyriproxyfen. Pyriproxyfen is an insecticide that interferes with the reproduction and development of female mosquitoes, effectively sterilising them [7].

The value to public health of DAI ITNs treated with both pyrethroids and insect growth regulators has not been established until recently. DAI ITNs may provide a solution to address vector pyrethroid resistance and may prove to have utility in future malaria control programmes. There is an urgent need to systematically review the evidence on the effectiveness of DAI ITNs as tools for the control and prevention of malaria.

This systematic review is specifically interested in two interventions. These interventions will be considered as separate review questions. The first intervention includes DAI ITNs treated with a pyrethroid and non-pyrethroid insecticide. The second intervention includes DAI ITNs treated with a pyrethroid and an insect growth regulator. The main objective of this review is to assess the benefits (on malaria transmission or burden) and harms (adverse effects and unintended consequences) of insecticidal nets treated with a pyrethroid and a second

active ingredient (either non-pyrethroid insecticide or insect growth regulator). Two review questions were formulated for this review, these questions are as follows:

1. In areas with ongoing malaria transmission, should insecticide-treated nets treated with a pyrethroid and non-pyrethroid insecticide versus either nets treated with pyrethroid insecticide alone or with pyrethroid insecticide in combination with Piperonyl butoxide (PBO) be used to prevent malaria in adults and children?

2. In areas with ongoing malaria transmission, should insecticide-treated nets treated with a pyrethroid and an insect growth regulator versus either nets treated with pyrethroid insecticide alone or with pyrethroid insecticide in combination with PBO be used to prevent malaria in adults and children?

## Methods

The methodology of this systematic review and meta-analyses is based on methods guidance from the Cochrane Handbook [8], JBI Manual for Evidence Synthesis [9], and the GRADE Working Group [10]. It has been reported in line with PRISMA 2020. The protocol was registered a priori on PROSPERO (registration number CRD42022333044) and has been published in F1000Research [11].

### Eligibility criteria

**Participants.** Studies conducted in adults and children who are residents of a region with ongoing malaria transmission and have been provided with an insecticide-treated net were eligible for this review.

**Interventions.** The interventions of interest are dual active ingredient (DAI) insecticide-treated nets (ITNs). DAI ITNs are eligible where they have been treated with a pyrethroid and non-pyrethroid insecticide (review question 1) or with a pyrethroid and an insect growth regulator (review question 2). The level of ITN distribution (per household or per individual) did not impact the eligibility of studies for inclusion in the review.

*Background interventions*. Studies conducted where background interventions were present were included if these background interventions were balanced between intervention and control arms.

**Comparators.** This systematic review considered studies that have compared the interventions of interest against nets treated with pyrethroid insecticide alone or with pyrethroid insecticide in combination with PBO. The same comparator(s) was used for both review questions specified above.

**Outcomes.** The following outcomes were considered for inclusion and are grouped into epidemiological outcomes, entomological outcomes, unintended benefits, and harms/ unintended consequences.

*Epidemiological*.

- Malaria case incidence rate–Defined as [malaria] symptoms plus [malaria] parasitaemia, over a population at risk or person-time. Detected either through passive or active surveillance.

- Malaria infection incidence–Defined as parasitaemia with or without symptoms, over a population at risk or person-time. Detected through passive or active surveillance.

- Incidence of severe disease–Defined as hospitalisation with parasitaemia, over a population at risk or person-time.

- Parasite prevalence–Parasitaemia with or without symptoms, over the population at risk for the specified duration. Detected through cross-sectional surveys.

- All-cause mortality–Number of deaths over the population at risk or person-time.

- Malaria mortality–Number of deaths attributed to malaria over the population at risk or person-time.

- Prevalence of anaemia–Defined by study thresholds of anaemia.

  *Entomological.*
  Studies containing data on entomological outcomes were only included in this review where data for epidemiological outcomes were also reported. These outcomes were only listed during data extraction and have not formed the basis of any outcome reporting.

- Entomological inoculation rate (EIR)–Defined as the number of infective bites received per person per unit of time.

- Sporozoite rate–Percentage of female *Anopheles* mosquitoes with sporozoites in the salivary glands.

- Anopheline density–Number of female anopheline mosquitoes in relation to the number of specified shelters or hosts or to a given period sampled, specifying the methods of collection.

- Biting rate–Average number of mosquito bites received by a host in a unit of time, specified according to the host and mosquito species.

- Mortality of adult female *Anopheles*–Defined as the mosquito being knocked down, immobile or unable to stand or take off for 24 hours after exposure to a discriminating concentration of an insecticide (or as reported in the primary evidence).

  *Contextual factors.*
  Studies containing data on contextual factors were only included in this review where epidemiological outcomes were also considered by the primary study.

- Values and preferences–The values and preferences of the individuals and populations receiving the intervention.

- Acceptability–Extent to which those receiving the intervention consider the intervention to be appropriate, based on anticipated or experienced cognitive and emotional responses to the intervention. Includes willingness to participate in the intervention.

- Health equity–Extent to which the intervention benefits all populations and the potential to discriminate based on sex, age, ethnicity, culture, language, sexual orientation or gender identity, disability status, education, socioeconomic status, residence, or any other characteristic.

- Financial and economic considerations–Costs, resource use, overall economic impact, cost-benefit, cost effectiveness.

- Feasibility considerations–legal barriers to implementation, programmatic considerations, timeliness (the ability to reach all targeted households/household members in a timely manner) etc.

  *Unintended benefits.*
  Epidemiological impact on other vector-borne diseases
  *Harms and/or unintended consequences of interventions.*

- Adverse effects known to be associated with insecticides, including skin irritation, irritation of upper airways, nausea, and headache.

- Human behaviour changes e.g., change in sleeping location.

- Any influence on neighbouring houses e.g., increased vector abundance/biting in houses without nets

- Environmental impacts such as biodiversity and ecosystem changes.

- Entomological impacts e.g., mosquito behaviour changes such as changes in outdoor biting rate, biting times, feeding preference, development of insecticide resistance, change in vector composition.

**Setting.**   Studies conducted in countries with ongoing malaria transmission were considered for this review. The presence of other background interventions did not impact on study eligibility if they were present in both arms equally. Studies where additional malaria interventions are considered standard of care were included if interventions (both malaria and non-malaria) were balanced between intervention and control arms.

**Study design.**   Only cluster randomised and non-randomised cluster-controlled studies that included more than one cluster per arm were considered for this review. Non-randomised controlled study designs were only considered for inclusion when there was a comparison/control group present. This could include historical controls. There were no exclusion rules based on any buffer period (i.e., when participants act as their own controls) or length of intervention or timing of measurement of outcomes. All observational studies and modelling studies were excluded.

Studies were not excluded based on language or publication status (i.e., published, unpublished, in press, in progress, pre-print). There were no date limitations. For studies published in languages other than English, Google Translate was to be used to determine whether the study meets inclusion criteria based on its title and abstract. Where studies were published in a language other than English and met the inclusion criteria, Google Translate translations were to be reviewed by a person fluent in that language.

## Search strategy

The literature search methods have been conducted in line with guidance from JBI [9] and Cochrane [8]. The search strategy aimed to locate both published and unpublished studies and was developed with the input of a medical librarian.

An initial limited search of PubMed via NCBI was undertaken to identify relevant articles on this topic. The terminology contained in the titles and abstracts of relevant articles, including related subject headings, were used to develop a full search strategy for malaria and insecticidal nets. The search strategy, including all identified keywords and subject headings, was adapted for each included database and/or information source, by using Polyglot [12] and with the aid of a medical librarian. No limits or filters were applied to the searches. The search strategies for each database were then peer-reviewed using the Peer Review of Electronic Search Strategies Guideline Statement [13]. The full search strategy for major databases is available in S1 in S1 File.

The databases searched included Cochrane Central Register of Controlled Trials (CENTRAL), published in The Cochrane Library (Wiley) and including the Cochrane Infectious Diseases Group Specialized Register; PubMed (NCBI); Embase (Ovid); CINAHL with Full Text (EBSCO), US National Institutes of Health Ongoing Trials Register (www.ClinicalTrials.

gov/); ISRCTN registry (www.isrctn.com/); The WHO's International Clinical Trials Registry Platform (WHO ICTRP (www.who.int.ictrp). Additionally, experts in the field and relevant organisations were asked whether they know of any studies (completed or ongoing) that are relevant to this review topic. The searches were run on June 7, 2022.

## Study selection and screening

Following the search, all identified citations were collated and uploaded into EndNote (Clarivate, Philadelphia, United States). Duplicates were removed using the Deduplicator [14]. The studies were then imported into Covidence (Veritas Health Innovation, Melbourne, Australia) where additional duplicates were identified and removed. Within Covidence, the studies were screened on their titles and abstracts by two independent reviewers (THB and SH) for assessment against the eligibility criteria for the review. Potentially relevant studies were retrieved in full. The full text of selected citations was then assessed in detail against the eligibility criteria by the same two independent reviewers. Studies that were excluded at full text screening as they did not meet the eligibility criteria have been recorded and the reasons for their exclusion reported (S2 in S1 File). Any disagreements between the two reviewers at each stage of the selection process were resolved through discussion.

## Data extraction

Data was extracted from papers included in the review by three independent reviewers (THB, SH, JCS*) using a tailored data extraction tool developed by the reviewers (S3 in S1 File). Any disagreements between the reviewers were resolved through discussion. The authors of one study protocol [15] were contacted directly for their data as the results of their work had not yet been published (discussed in detail in the results).

## Assessment of risk of bias

Three review authors (THB, SH, JCS*) assessed the risk of bias for each study using the Cochrane Risk of Bias 2 tool for cluster-randomised trials [16]. The domains of bias considered in this tool include bias arising from the randomisation process, bias arising from the timing of identification or recruitment of participants, bias due to deviations from the intended interventions, bias due to missing outcome data, bias in measurement of the outcome and bias in selection of the reported result. All risk of bias assessment was undertaken at the result level. Any disagreements between the reviewers in assessing the risk of bias were resolved through discussion.

## Data synthesis and meta-analysis

Where possible, epidemiological outcomes were pooled using pair-wise meta-analysis in Review Manager 5 (RevMan5). Results have been pooled when data for the same outcome has been reported between studies and according to the active-ingredient composition of the DAI ITN intervention and of the pyrethroid-only ITN or pyrethroid-PBO ITN comparators. Data were also pooled at time-points measured in the contributing studies (6-months, 12-months, 18-months, 24-months post intervention). As some studies provided data for up to 18-months post-intervention and some provided data for up to 24-months post-intervention, these outcome results have been combined under the classification of 'furthest possible follow-up'. Also included in this classification is data derived from stepped-wedge trials. Where only one study had contributed data to a particular outcome, a forest plot was presented for illustrative purposes. A narrative description of the results has been presented alongside the meta-analysis.

Where outcome data between studies cannot be pooled together in a meta-analysis, a narrative synthesis has been presented.

For dichotomous data, effect sizes have been presented as odds ratios. These results have been presented with their 95% confidence intervals (CIs). Where incidence rates were reported, incidence rate ratios have been reported with their 95% CIs in the meta-analysis. Calculation of 95% CIs took account of the clustered nature of the data where appropriate. When three or more studies contribute to a meta-analysis, a random effects model has been used. A fixed effect model was used when there are only two studies contributing to a meta-analysis. Cost data and data related to contextual factors have been narratively synthesised. Entomological outcomes listed in the included studies have been reported in S3 in S1 File.

## Assessment of heterogeneity and publication bias

Heterogeneity (both clinical and methodological) was first assessed by comparing the included studies against each other in terms of the eligibility criteria specified above. Statistical heterogeneity was assessed through visual inspection of the forest plot and by the Cochran's Q (P value 0.05), and $I^2$ statistic. Interpretation of the $I^2$ statistic was according to the guidance in the Cochrane Handbook for Systematic Reviews of Interventions [8] and occurred as follows:

- 0% to 40%: heterogeneity might not be important;

- 30% to 60%: may represent moderate heterogeneity;

- 50% to 90%: may represent substantial heterogeneity; or

- 75% to 100%: considerable heterogeneity.

The typical statistical tests of publication bias were not appropriate [17, 18] as fewer than 10 studies were included in all meta-analyses. Efforts were made to reduce the impact of publication bias in this review by seeking both published and unpublished literature using the comprehensive search strategy discussed above and provided in S1 in S1 File.

## Subgroup and sensitivity analysis

Where the data was available, several potential effect modifiers were assessed through subgroup analyses. These included:

- Insecticides used for both active ingredients and manufacturer.

- Malaria vector species.

- Setting (Urbanicity, classed as rural/ urban/ peri-urban).

Subgroups were assessed on their credibility of being a genuine effect modifier using the Instrument for assessing the Credibility of Effect Modification (ICEMAN) [19]. This is a tool that reviewers can use based on answering a series of questions that address specific criteria that can be used to evaluate whether an effect modification is likely [19]. ICEMAN credibility assessment statements are expressed as very low (very likely no effect modification), low (likely no effect modification), moderate (likely effect modification), and high (very likely effect modification).

## GRADE

The GRADE approach [10] for grading the certainty of evidence was followed. GRADE Evidence Profiles were created using "GRADEpro GDT" for each comparison considered. The

evidence profiles have presented the following information for each outcome: absolute risks for the treatment and control, estimates of relative risk, and a rating of the certainty of the evidence based. As all evidence has been derived from RCTs, the certainty of evidence has started as high and has been downgraded appropriately. All instances of downgrading have been documented in the footnotes in the summary of findings tables (Tables 1–4). The following outcomes have been presented in the summary of findings tables (where applicable):

- Malaria case incidence rate (overall)

- Malaria case incidence rate (1-year post intervention)

- Malaria case incidence rate (2-years post intervention)

- Parasite prevalence (6-months follow-up)

- Parasite prevalence (12-months follow-up)

- Parasite prevalence (18-months follow-up)

- Parasite prevalence (24-months follow-up)

## Results

### Results of the search

There were 8998 citation records identified in the initial database search (i.e., PubMed, Embase, CINAHL, Cochrane Library) and 496 citation records were identified from the trial registries (ClinicalTrials.gov, WHO ICTRP, ISRCTN), for a total of 9494 citation records. Of these, a total of 3694 records were removed (3662 records were identified and removed via the Deduplicator [14] and a further 32 records were identified and removed via Covidence). This left 5800 unique citation records to be screened. Two citation records were identified through direct correspondence with the authors (described below) and through manual searching through the ClinicalTrials.gov trial registry. The former of these records was an ongoing trial (NCT04566510) which has been noted for future reviews on this topic but has not contributed to any of the analysis of this report.

The records were screened by title and abstract and 5752 citations were excluded for not meeting the inclusion criteria. This left 48 records in which the reports were sought and were screened at the full-text level. There were 36 reports that were excluded for not meeting the inclusion criteria. Of these 36, 21 reports were excluded for having an ineligible study design, eight reports were excluded for having ineligible outcomes and seven reports were excluded for having ineligible interventions.

The 12 remaining reports were then merged at the study level, leaving three studies (12 reports) to be included in the review. The report that was identified through direct correspondence with the authors was an ongoing study that had been accepted for publication but was still in production (as of writing, this study has been published by the Lancet). This study has been merged with the reports of the protocol that were identified during the search and screening procedures. Therefore, the final totals were three studies included in this review which have been reported in 12 reports identified through the search and one report identified via direct correspondence with the study authors. The breakdown of reports to studies is presented in Table 5.

The PRISMA flow diagram of this screening process is presented below (Fig 1).

### Characteristics of the included studies

**Study designs and time periods.** This review has included three studies [6, 20, 21], one of which was a trial [15] that was only recently accepted for publication [20] following peer-

**Table 1. Evidence profile: Chlorfenapyr-pyrethroid ITNs compared to pyrethroid-only ITNs for prevention of malaria.**

| Certainty assessment | | | | | | | Summary of findings | | | | |
|---|---|---|---|---|---|---|---|---|---|---|---|
| Participants (studies) Follow-up | Risk of bias | Inconsistency | Indirectness | Imprecision | Publication bias | Overall certainty of evidence | Study event rates (%) | | Relative effect (95% CI) | Anticipated absolute effects | |
| | | | | | | | With Pyrethroid-only nets | With Chlorfenapyr-pyrethroid nets | | Risk with Pyrethroid-only nets | Risk difference with Chlorfenapyr-pyrethroid nets |
| **Malaria Case Incidence (overall)** | | | | | | | | | | | |
| 2000-person years (2 RCTs) Length of time observed: <1 month to 24 months Based on data from at least 61,183 participants (participant numbers unavailable in 1 study) | not serious | not serious | not serious | not serious | none | ⊕⊕⊕⊕ High[a,b] | 677.58 cases over 1000-person years (67.8%) | 355.44 cases over 1000-person years (35.5%) | **Incidence Rate Ratio 0.55** (0.44 to 0.68)[c] | 678 cases per 1,000-person years | **305 fewer cases per 1,000-person years** (from 380 fewer cases to 216 fewer cases)[d] |
| **Malaria Case Incidence (1-year post-intervention)** | | | | | | | | | | | |
| 2000-person years (2 RCTs) Length of time observed: <1 month to 12 months Based on data from at least 61,183 participants (participant numbers unavailable in 1 study) | not serious | not serious | not serious | not serious | none | ⊕⊕⊕⊕ High[e,f] | 485.52 cases over 1000-person years (48.5%) | 213.01 cases over 1000-person years (21.3%) | **Incidence Rate Ratio 0.47** (0.35 to 0.63)[c] | 486 cases per 1,000-person years | **257 fewer cases per 1,000-person years** (from 315 fewer cases to 180 fewer cases)[d] |
| **Malaria Case Incidence (2-years post-intervention)** | | | | | | | | | | | |
| 2000 (2 RCTs) Length of time observed: 12 months to 24 months Based on data from at least 61,183 participants (participant numbers unavailable in 1 study) | not serious | not serious | not serious | not serious | none | ⊕⊕⊕⊕ High[g,h] | 815.38 cases over 1000-person years (81.5%) | 465.12 cases over 1000-person years (46.5%) | **Incidence Rate Ratio 0.67** (0.61 to 0.75)[c] | 815 cases per 1,000-person years | **269 fewer cases per 1,000-person years** (from 318 fewer cases to 204 fewer cases)[c] |
| **Parasite Prevalence (6-months follow-up)** | | | | | | | | | | | |
| 2249 (1 RCT) | not serious | not serious | not serious | not serious | none | ⊕⊕⊕⊕ High | 412/1471 (28%) | 231/1475 (15.6%) | **OR 0.47** (0.32 to 0.69)[i] | 312 per 1,000 | **165 fewer per 1,000** (from 212 fewer to 97 fewer) |
| **Parasite Prevalence (12-months follow-up)** | | | | | | | | | | | |
| 2473 (1 RCT) | not serious | not serious | not serious | not serious | none | ⊕⊕⊕⊕ High | 642/1227 (52.3%) | 509/1246 (40.9%) | **OR 0.47** (0.31 to 0.71)[i] | 523 per 1,000 | **277 fewer per 1,000** (from 361 fewer to 152 fewer) |
| **Parasite Prevalence (18-months follow-up)** | | | | | | | | | | | |
| 5445 (2 RCTs) | not serious | not serious | not serious | not serious | none | ⊕⊕⊕⊕ High[j,k] | 1218/2716 (44.8%) | 923/2729 (33.8%) | **OR 0.63** (0.49 to 0.80)[i] | 448 per 1,000 | **166 fewer per 1,000** (from 228.48 fewer to 90 fewer) |

(Continued)

**Table 1.** (Continued)

| Certainty assessment | | | | | | | Summary of findings | | Relative effect (95% CI) | Anticipated absolute effects | |
|---|---|---|---|---|---|---|---|---|---|---|---|
| Participants (studies) Follow-up | Risk of bias | Inconsistency | Indirectness | Imprecision | Publication bias | Overall certainty of evidence | Study event rates (%) | | | | |
| | | | | | | | With Pyrethroid-only nets | With Chlorfenapyr-pyrethroid nets | | Risk with Pyrethroid-only nets | Risk difference with Chlorfenapyr-pyrethroid nets |
| **Parasite Prevalence (24-months follow-up)** | | | | | | | | | | | |
| 2471 (1 RCT) | not serious | not serious | not serious | not serious | none | ⊕⊕⊕⊕ High | 549/1199 (45.8%) | 326/1272 (25.6%) | OR 0.45 (0.30 to 0.68)[i] | 458 per 1,000 | **252 fewer per 1,000** (from 321 fewer to 146 fewer) |

**CI**: confidence interval; **RR**: risk ratio

Explanations

a. ICEMAN effect modifier credibility assessments were conducted on this analysis for vector (An. Funestus versus *An. Gambiae sensu stricto* and *An. Coluzzii*). Test for subgroup differences between subgroups was p = 0.02 (chance a very likely explanation). ICEMAN credibility assessment determined very low credibility, very likely no effect modification. Of note the ICEMAN tool is not intended to rule out a true difference. As such, only the overall effect is presented however uncertainty remains. Forest plot and ICEMAN assessment presented in S1 File.

b. ICEMAN effect modifier credibility assessments were conducted on this analysis for setting (Rural versus Mixed). Test for subgroup differences between subgroups was p = 0.057 (chance a very likely explanation). ICEMAN credibility assessment determined very low credibility, very likely no effect modification. Of note the ICEMAN tool is not intended to rule out a true difference. As such, only the overall effect is presented however uncertainty remains. Forest plot and ICEMAN assessment presented in S1 File.

c. Adjusted Incidence Rate Ratio

d. Absolute calculation performed manually using unadjusted data, as GRADEPro cannot calculate using IRR

e. ICEMAN effect modifier credibility assessments were conducted on this analysis for vector (*An. Funestus* versus *An. Gambiae sensu stricto* and *An. Coluzzii*). Test for subgroup differences between subgroups was p = 0.88 (chance a very likely explanation). ICEMAN credibility assessment determined very low credibility, very likely no effect modification. Of note the ICEMAN tool is not intended to rule out a true difference. As such, only the overall effect is presented however uncertainty remains. Forest plot and ICEMAN assessment presented in S1 File.

f. ICEMAN effect modifier credibility assessments were conducted on this analysis for setting (Rural versus Mixed). Test for subgroup differences between subgroups was p = 0.88 (chance a very likely explanation). ICEMAN credibility assessment determined very low credibility, very likely no effect modification. Of note the ICEMAN tool is not intended to rule out a true difference. As such, only the overall effect is presented however uncertainty remains. Forest plot and ICEMAN assessment presented in S1 File.

g. ICEMAN effect modifier credibility assessments were conducted on this analysis for vector (*An. Funestus* versus *An. Gambiae sensu stricto* and *An. Coluzzii*). Test for subgroup differences between subgroups was p = 0.05 (chance a very likely explanation). ICEMAN credibility assessment determined very low credibility, very likely no effect modification. Of note the ICEMAN tool is not intended to rule out a true difference. As such, only the overall effect is presented however uncertainty remains. Forest plot and ICEMAN assessment presented in S1 File.

h. ICEMAN effect modifier credibility assessments were conducted on this analysis for setting (Rural versus Mixed). Test for subgroup differences between subgroups was p = 0.05 (chance a very likely explanation). ICEMAN credibility assessment determined very low credibility, very likely no effect modification. Of note the ICEMAN tool is not intended to rule out a true difference. As such, only the overall effect is presented, however uncertainty remains. Forest plot and ICEMAN assessment presented in S1 File.

i. Adjusted Odds Ratio

j. ICEMAN effect modifier credibility assessments were conducted on this analysis for vector (*An. Funestus* versus *An. Gambiae sensu stricto* and *An. Coluzzii*). Test for subgroup differences between subgroups was p = 0.71 (chance a very likely explanation). ICEMAN credibility assessment determined very low credibility, very likely no effect modification. Of note the ICEMAN tool is not intended to rule out a true difference. As such, only the overall effect is presented however uncertainty remains. Forest plot and ICEMAN assessment presented in S1 File.

k. ICEMAN effect modifier credibility assessments were conducted on this analysis for setting (Rural versus Mixed). Test for subgroup differences between subgroups was p = 0.71 (chance a very likely explanation). ICEMAN credibility assessment determined very low credibility, very likely no effect modification. Of note the ICEMAN tool is not intended to rule out a true difference. As such, only the overall effect is presented however uncertainty remains. Forest plot and ICEMAN assessment presented in S1 File.

**Table 2. Evidence profile: Chlorfenapyr-pyrethroid nets compared to pyrethroid-PBO nets for prevention of malaria.**

| Certainty assessment | | | | | | | Summary of findings | | | Anticipated absolute effects | |
|---|---|---|---|---|---|---|---|---|---|---|---|
| | | | | | | | Study event rates (%) | | Relative effect (95% CI) | | |
| Participants (studies) Follow-up | Risk of bias | Inconsistency | Indirectness | Imprecision | Publication bias | Overall certainty of evidence | With Pyrethroid-PBO nets | With Chlorfenapyr-pyrethroid nets | | Risk with Pyrethroid-PBO nets | Risk difference with Chlorfenapyr-pyrethroid nets |
| **Malaria Case Incidence (overall)** | | | | | | | | | | | |
| 2000-person years (1 RCT) Length of time observed: <1 month to 24 months Based on data from at least 61,183 participants (participant numbers unavailable in 1 study) | serious[a] | not serious | not serious | not serious | none | ⊕⊕⊕◯ Moderate | 333.16 cases over 1000-person years (33.3%) | 227.34 cases over 1000-person years (22.7%) | **Incidence Rate Ratio 0.68** (0.59 to 0.79) | 333 cases per 1,000-person years | **107 fewer cases per 1,000-peson years** (from 137 fewer cases to 70 fewer cases)[b] |
| **Malaria Case Incidence (1-year post-intervention)** | | | | | | | | | | | |
| 2000-person years (1 RCT) Length of time observed: <1 month to 12 months Based on data from at least 61,183 participants (participant numbers unavailable in 1 study) | serious[a] | not serious | not serious | very serious[c] | none | ⊕◯◯◯ Very Low | 133.49 cases over 1000-person years (13.3%) | 130.84 cases over 1000-person years (13.1%) | **Incidence Rate Ratio 0.98** (0.71 to 1.36) | 133 cases per 1,000-person years | **3 fewer cases per 1,000-person years** (from 39 fewer cases to 48 more cases)[b] |
| **Malaria Case Incidence (2-years post-intervention)** | | | | | | | | | | | |
| 2000-person years (1 RCT) Length of time observed: 12 months to 24 months Based on data from at least 61,183 participants (participant numbers unavailable in 1 study) | serious[a] | not serious | not serious | not serious | none | ⊕⊕⊕◯ Moderate | 482.95 cases over 1000-person years (48.3%) | 314.95 cases over 1000-person years (31.5%) | **Incidence Rate Ratio 0.65** (0.55 to 0.77) | 483 cases per 1,000-person years | **155 fewer cases per 1,000-person years** (from 198 fewer cases to 101 fewer cases)[b] |
| **Parasite Prevalence (12-months follow-up)** | | | | | | | | | | | |
| 2197 (1 RCT) | serious[a] | not serious | not serious | serious[d] | none | ⊕⊕◯◯ Low | 206/1071 (19.2%) | 176/1126 (15.6%) | **OR 0.78** (0.62 to 0.97) | 192 per 1,000 | **42 fewer per 1,000** (from 73 fewer to 6 fewer) |

*(Continued)*

**Table 2.** (Continued)

| Certainty assessment | | | | | | | Summary of findings | | | Anticipated absolute effects | |
|---|---|---|---|---|---|---|---|---|---|---|---|
| | | | | | | | Study event rates (%) | | Relative effect (95% CI) | | |
| Participants (studies) Follow-up | Risk of bias | Inconsistency | Indirectness | Imprecision | Publication bias | Overall certainty of evidence | With Pyrethroid-PBO nets | With Chlorfenapyr-pyrethroid nets | | Risk with Pyrethroid-PBO nets | Risk difference with Chlorfenapyr-pyrethroid nets |
| **Parasite Prevalence (18-months follow up)** | | | | | | | | | | | |
| 2406 (1 RCT) | serious[a] | not serious | not serious | serious[e] | none | ⊕⊕○○ Low | 502/1160 (43.3%) | 509/1246 (40.9%) | **OR 0.91** (0.77 to 1.04) | 433 per 1,000 | **39 fewer per 1,000** (from 100 fewer to 17 more) |
| **Parasite Prevalence (24-months follow-up)** | | | | | | | | | | | |
| 2531 (1 RCT) | serious[a] | not serious | not serious | not serious | none | ⊕⊕⊕○ Moderate | 512/1259 (40.7%) | 326/1272 (25.6%) | **OR 0.50** (0.42 to 0.60) | 407 per 1,000 | **203 fewer per 1,000** (from 236 fewer to 163 fewer) |

**CI:** confidence interval; **RR:** risk ratio

Explanations

a. Only unadjusted data was available for use for this comparison, and therefore there are serious issues with risk of bias.

b. Absolute calculation performed manually as GRADEPro cannot calculate using IRR.

c. Confidence intervals are very wide (39 fewer to 48 more) and may have crossed many important decision-making threshold (including line of no effect).

d. Confidence intervals are wide (73 fewer to 6 fewer) and may have crossed many important decision-making thresholds.

e. Confidence intervals are wide (from 100 fewer to 17 more) and may have crossed many important decision-making threshold (including line of no effect).

**Table 3. Evidence profile: Pyriproxyfen-pyrethroid nets compared to pyrethroid-only nets for prevention of malaria.**

| Certainty assessment | | | | | | | Summary of findings | | | Anticipated absolute effects | |
|---|---|---|---|---|---|---|---|---|---|---|---|
| | | | | | | | Study event rates (%) | | Relative effect (95% CI) | | |
| Participants (studies) Follow-up | Risk of bias | Inconsistency | Indirectness | Imprecision | Publication bias | Overall certainty of evidence | With Pyrethroid-only nets | With Pyriproxyfen-pyrethroid nets | | Risk with Pyrethroid-only nets | Risk difference with Pyriproxyfen-pyrethroid nets |
| **Malaria Case Incidence (overall)** | | | | | | | | | | | |
| 2000-person years (3 RCTs) Length of time observed: 5 months to 24 months Based on data from at least 63,163 participants (participant numbers unavailable in 1 study) | not serious | not serious | not serious | very serious[a] | none | ⊕⊕◯◯ Low[b,c,d] | 1036.93 cases over 1000-person years (103.7%) | 929.22 cases over 1000-person years (92.9%) | Incidence Rate Ratio **0.90** (0.73 to 1.13)[e] | 1,037 cases per 1,000-person years | **104 fewer cases per 1,000-person years** (from 280 fewer cases to 135 more cases)[f] |
| **Malaria Case Incidence (1-year post-intervention)** | | | | | | | | | | | |
| 2000-person years (2 RCTs) Length of time observed: <1 month to 12 months Based on data from at least 61,183 participants (participant numbers unavailable in 1 study) | not serious | not serious | not serious | not serious | none | ⊕⊕⊕⊕ High[g,h] | 485.52 cases over 1000-person years (48.6%) | 392.74 cases over 1000-person years (39.3%) | Incidence Rate Ratio **0.66** (0.47 to 0.85)[e] | 487 cases per 1,000-person years | **166 fewer cases per 1,000-person years** (from 258 fewer cases to 73 fewer cases)[f] |
| **Malaria Case Incidence (2-year post-intervention)** | | | | | | | | | | | |
| 2000 (2 RCTs) Length of time observed: 12 months to 24 months Based on data from at least 61,183 participants (participant numbers unavailable in 1 study) | not serious | not serious[i] | not serious | very serious[j] | none | ⊕⊕⊕◯ Moderate[k,l] | 815.39 cases over 1000-person years (81.5%) | 715.84 cases over 1000-person years (71.6%) | Incidence Rate Ratio **0.94** (0.75 to 1.17)[e] | 815 cases per 1,000-person years | **49 fewer cases per 1,000** (from 204 fewer to 138 more)[f] |
| **Parasite Prevalence (6-months follow-up)** | | | | | | | | | | | |
| 2934 (1 RCT) | not serious | not serious | not serious | serious[m] | none | ⊕⊕⊕◯ Moderate | 412/1471 (28.0%) | 394/1463 (26.9%) | **OR 0.92** (0.63 to 1.34)[n] | 280 per 1,000 | **22 fewer per 1,000** (from 104 fewer to 95 more) |
| **Parasite Prevalence (12-months follow-up)** | | | | | | | | | | | |
| 2192 (1 RCT) | not serious | not serious | not serious | serious[o] | none | ⊕⊕⊕◯ Moderate | 350/1123 (31.2%) | 232/1069 (21.7%) | **OR 0.69** (0.46 to 1.04)[n] | 96 per 1,000 | **93 fewer per 1,000** (from 168 fewer to 12 more) |
| **Parasite Prevalence (18-months follow-up)** | | | | | | | | | | | |
| 5337 (2 RCTs) | not serious | not serious | not serious | very serious[p] | none | ⊕⊕◯◯ Low[q,r] | 1218/2716 (44.8%) | 1147/2631 (43.8%) | **OR 0.97** (0.76 to 1.26)[n] | 448 per 1,000 | **13 fewer per 1,000** (from 108 fewer to 116 more) |
| **Parasite Prevalence (24-months follow-up)** | | | | | | | | | | | |

(*Continued*)

**Table 3.** (Continued)

| Certainty assessment | | | | | | | Summary of findings | | | | |
|---|---|---|---|---|---|---|---|---|---|---|---|
| Participants (studies) Follow-up | Risk of bias | Inconsistency | Indirectness | Imprecision | Publication bias | Overall certainty of evidence | Study event rates (%) | | Relative effect (95% CI) | Anticipated absolute effects | |
| | | | | | | | With Pyrethroid-only nets | With Pyriproxyfen-pyrethroid nets | | Risk with Pyrethroid-only nets | Risk difference with Pyriproxyfen-pyrethroid nets |
| 2457 (1 RCT) | not serious | not serious | not serious | serious[s] | none | ⊕⊕⊕◯ Moderate | 549/1199 (45.8%) | 472/1258 (37.5%) | OR 0.77 (0.54 to 1.16)[n] | 458 per 1,000 | **105 fewer per 1,000** (from 192 fewer to 13 more) |

**CI:** confidence interval; **RR:** risk ratio

Explanations

a. Confidence intervals are very wide (from 280 fewer to 135 more) and may have crossed many important decision-making threshold (including line of no effect).

b. ICEMAN effect modifier credibility assessments were conducted on this analysis for net type (Royal Guard versus Sumitomo Chemical). Test for subgroup differences between subgroups was p = 0.89 (chance a very likely explanation). ICEMAN credibility assessment determined very low credibility, very likely no effect modification. Of note the ICEMAN tool is not intended to rule out a true difference. As such, only the overall effect is presented however uncertainty remains. Forest plot and ICEMAN assessment presented in S1 File.

c. ICEMAN effect modifier credibility assessments were conducted on this analysis for vector (*An. funestus* versus *An. gambiae sensu stricto* and *An. coluzzii*. Test for subgroup differences between subgroups was p = 0.20 (chance a very likely explanation). ICEMAN credibility assessment determined very low credibility, very likely no effect modification. Of note the ICEMAN tool is not intended to rule out a true difference. As such, only the overall effect is presented however uncertainty remains. Forest plot and ICEMAN assessment presented in S1 File.

d. ICEMAN effect modifier credibility assessments were conducted on this analysis for setting (Rural versus Mixed). Test for subgroup differences between subgroups was p = 0.57 (chance a very likely explanation). ICEMAN credibility assessment determined very low credibility, very likely no effect modification. Of note the ICEMAN tool is not intended to rule out a true difference. As such, only the overall effect is presented however uncertainty remains. Forest plot and ICEMAN assessment presented in S1 File.

e. Adjusted Incidence Rate Ratio

f. Absolute calculation performed manually as GRADEPro cannot calculate using IRR

g. ICEMAN effect modifier credibility assessments were conducted on this analysis for vector (*An. funestus* versus *An. gambiae sensu stricto* and *An. coluzzii*). Test for subgroup differences between subgroups was p = 0.35 (chance a very likely explanation). ICEMAN credibility assessment determined low credibility, likely no effect modification. Of note the ICEMAN tool is not intended to rule out a true difference. As such, only the overall effect is presented however uncertainty remains. Forest plot and ICEMAN assessment presented in S1 File.

h. ICEMAN effect modifier credibility assessments were conducted on this analysis for setting (Rural versus Mixed). For each subgroup the IRR was: Rural = 0.79 (0.66, 0.95); Mixed = 0.83 (0.67, 1.03). Test for subgroup differences between subgroups was p = 0.94 (chance a very likely explanation). ICEMAN credibility assessment determined very low credibility, very likely no effect modification. Of note the ICEMAN tool is not intended to rule out a true difference. As such, only the overall effect is presented however uncertainty remains. Forest plot and ICEMAN assessment presented in S1 File.

i. Borderline inconsistency, point estimate vary to some extent and I2 48%. However, not deemed serious enough to rate down.

j. Confidence intervals are very wide (from 204 fewer to 138 more) and may have crossed an important decision-making threshold (line of no effect).

k. ICEMAN effect modifier credibility assessments were conducted on this analysis for vector (*An. funestus* versus *An. gambiae sensu stricto* and *An. coluzzii*). Test for subgroup differences between subgroups was p = 0.97 (chance a very likely explanation). ICEMAN credibility assessment determined very low credibility, very likely no effect modification. Of note the ICEMAN tool is not intended to rule out a true difference. As such, only the overall effect is presented however uncertainty remains. Forest plot and ICEMAN assessment presented in S1 File.

l. ICEMAN effect modifier credibility assessments were conducted on this analysis for setting (Rural versus Mixed). Test for subgroup differences between subgroups was p = 0.71 (chance a very likely explanation). ICEMAN credibility assessment determined very low credibility, very likely no effect modification. Of note the ICEMAN tool is not intended to rule out a true difference. As such, only the overall effect is presented however uncertainty remains. Forest plot and ICEMAN assessment presented in S1 File.

m. Confidence intervals are wide (from 103 fewer to 95 more) and may have crossed many important decision-making thresholds (including line of no effect).

n. Adjusted Odds Ratio

o. Confidence intervals are wide (from 168 fewer to 12 more) and may have crossed many important decision-making thresholds (including line of no effect).

p. Confidence intervals are very wide (from 206 fewer to 72 more) and may have crossed many important decision-making thresholds (including line of no effect).

q. ICEMAN effect modifier credibility assessments were conducted on this analysis for vector (*An. funestus* versus *An. gambiae sensu stricto* and *An. coluzzii*).Test for subgroup differences between subgroups was p = 0.50 (chance a very likely explanation). ICEMAN credibility assessment determined very low credibility, very likely no effect modification. Of note the ICEMAN tool is not intended to rule out a true difference. As such, only the overall effect is presented however uncertainty remains. Forest plot and ICEMAN assessment presented in S1 File.

r. ICEMAN effect modifier credibility assessments were conducted on this analysis for setting (Rural versus Mixed). Test for subgroup differences between subgroups was p = 0.28 (chance may not explain). ICEMAN credibility assessment determined low credibility, likely no effect modification. Of note the ICEMAN tool is not intended to rule out a true difference. As such, only the overall effect is presented however uncertainty remains. Forest plot and ICEMAN assessment presented in S1 File.

s. Confidence intervals are wide (from 192 fewer to 13 more) and may have crossed many important decision-making thresholds (including line of no effect).

**Table 4. Evidence profile: Pyriproxyfen-pyrethroid nets compared to pyrethroid-PBO nets for prevention of malaria.**

| Certainty assessment | | | | | | | Summary of findings | | | Anticipated absolute effects | |
|---|---|---|---|---|---|---|---|---|---|---|---|
| Participants (studies) Follow-up | Risk of bias | Inconsistency | Indirectness | Imprecision | Publication bias | Overall certainty of evidence | Study event rates (%) | | Relative effect (95% CI) | | |
| | | | | | | | With Pyrethroid-PBO nets | With Pyriproxyfen-pyrethroid nets | | Risk with Pyrethroid-PBO nets | Risk difference with Pyriproxyfen-pyrethroid nets |
| **Malaria Case Incidence (overall)** | | | | | | | | | | | |
| 2000-person years (1 RCT) Length of time observed: <1 month to 24 months Based on data from at least 61,183 participants (participant numbers unavailable in 1 study) | serious[a] | not serious | not serious | not serious | none | ⊕⊕⊕◯ Moderate | 333.16 cases over 1000-person years (33.3%) | 415.98 cases over 1000-person years (41.6%) | **Incidence Rate Ratio 1.25** (1.10 to 1.41) | 333 cases per 1,000-person years | **83 more cases per 1,000-person years** (from 33 more cases to 137 more cases)[b] |
| **Malaria Case Incidence (1-year post-intervention)** | | | | | | | | | | | |
| 2000-person years (1 RCT) Length of time observed: <1 month to 12 months Based on data from at least 61,183 participants (participant numbers unavailable in 1 study) | serious[a] | not serious | not serious | not serious | none | ⊕⊕⊕◯ Moderate | 130.84 cases over 1000-person years (13.1%) | 266.33 cases over 1000-person years (26.6%) | **Incidence Rate Ratio 2.04** (1.55 to 2.68) | 131 cases per 1,000-person years | **136 more cases per 1,000-person years** (from 72 more cases to 220 more cases)[b] |
| **Malaria Case Incidence (2-years post-intervention)** | | | | | | | | | | | |
| 2000-person years (1 RCT) Length of time observed: 12 months to 24 months Based on data from at least 61,183 participants (participant numbers unavailable in 1 study) | serious[a] | not serious | not serious | serious[c] | none | ⊕⊕◯◯ Low | 482.95 cases over 1000-person years (48.3%) | 530.86 cases over 1000-person years (53.1%) | **Incidence Rate Ratio 1.10** (0.95 to 1.27) | 483 cases per 1,000-person years | **48 more cases per 1,000-peson years** (from 24 fewer cases to 130 more cases)[b] |
| **Parasite Prevalence (12-months follow-up)** | | | | | | | | | | | |
| 2140 (1 RCT) | serious[a] | not serious | not serious | serious[d] | none | ⊕⊕◯◯ Low | 206/1071 (19.2%) | 232/1,069 (21.7%) | **OR 1.16** (0.94 to 1.44) | 192 per 1,000 | **31 more per 1,000** (from 11 fewer to 84 more) |
| **Parasite Prevalence (18-months follow-up))** | | | | | | | | | | | |

(*Continued*)

**Table 4.** (Continued)

| Certainty assessment | | | | | | | Summary of findings | | | Anticipated absolute effects | |
|---|---|---|---|---|---|---|---|---|---|---|---|
| Participants (studies) Follow-up | Risk of bias | Inconsistency | Indirectness | Imprecision | Publication bias | Overall certainty of evidence | Study event rates (%) With Pyrethroid-PBO nets | With Pyriproxyfen-pyrethroid nets | Relative effect (95% CI) | Risk with Pyrethroid-PBO nets | Risk difference with Pyriproxyfen-pyrethroid nets |
| 2313 (1 RCT) | serious[a] | not serious | not serious | not serious | none | ⊕⊕⊕◯ Moderate | 502/1160 (43.3%) | 583/1,153 (50.6%) | **OR 1.34** (1.14 to 1.58) | 433 per 1,000 | **147 more per 1,000** (from 61 more to 251 more) |
| **Parasite Prevalence (24-months follow-up)** | | | | | | | | | | | |
| 2517 (1 RCT) | serious[a] | not serious | not serious | serious[e] | none | ⊕⊕◯◯ Low | 512/1259 (40.7%) | 472/1,258 (37.5%) | **OR 0.88** (0.75 to 1.03) | 407 per 1,000 | **49 fewer per 1,000** (from 102 fewer to 12 more) |

**CI:** confidence interval; **OR:** odds ratio; **RR:** risk ratio

Explanations

a. Only unadjusted data was available for use for this comparison, and therefore there are serious issues with risk of bias.

b. Absolute calculation performed manually as GRADEPro cannot calculate using IRR

c. Confidence intervals are very wide (from 24 fewer 130 more) and may have crossed many important decision-making thresholds (including line of no effect).

d. Confidence intervals are wide (from 11 fewer to 84 more) and may have crossed many important decision-making thresholds (including line of no effect).

e. Confidence intervals are wide (from 102 fewer to 12 more) and may have crossed many important decision-making thresholds (including line of no effect).

**Table 5. Breakdown of reports to studies included in the systematic review.**

| Study Citation | Number of Reports |
| --- | --- |
| Accrombessi, Cook [15, 20] | 3 |
| | 2 identified through screening |
| | 1 identified through direct author correspondence with authors |
| Mosha, Kulkarni [6] | 3 |
| Tiono, Ouédraogo [21] | 7 |

review. All the included studies were cluster-randomised control trials, with the study from Tiono, Ouédraogo [21] employing a stepped-wedge design for intervention implementation. The years during which the trials took place were between 2014–2015 [21], 2018–2020 [6], and 2020–2022 [20].

**Population, setting and vector characteristics.** The sample size ranged from approximately 4,000 households [20] to 39,307 households [6]. The number of participants for each study ranged from 1,980 [21] to 61,183 [6]. Studies included both adults and children in their design, however children were prioritised in the measurement of the outcomes and population demographics (data from adults included in select outcomes, detailed below). Accrombessi, Cook [20] reported data for adults and children (collected from cross-sectional studies) and only children (active-case detection, details below) between the ages of 6 months to 9 years old

**PRISMA 2020 flow diagram for new systematic reviews which included searches of databases, registers and other sources**

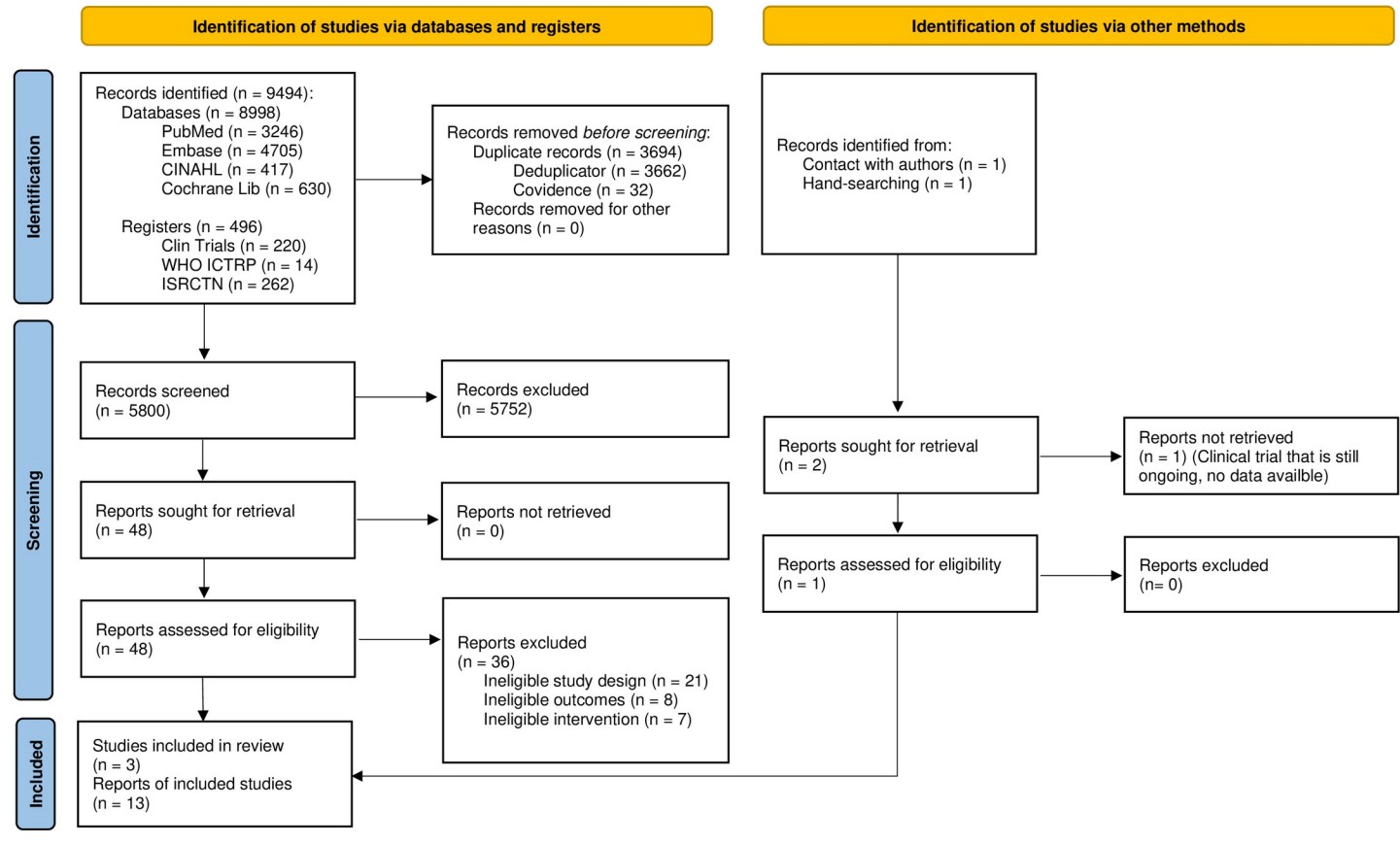

**Fig 1. PRISMA flow diagram [24].**

who did not have severe illnesses and resided in the study villages at the time of the intervention. Mosha, Kulkarni [6] included households with at least one child of appropriate age (between six months to ten years old) who permanently lived in households recruited through a census. Adults were also considered in the data from cross-sectional surveys (details below). Tiono, Ouédraogo [21] (2018) included children selected randomly from a census, who were between the ages of 6 months to 5 years old. The percentage of female to male children was balanced for all included studies at 48% [20], 51.7% [6] and 49% [21].

The countries involved in this review included Benin [20], Burkina Faso [21] and Tanzania [6] with the trials in Burkina Faso [21] and Tanzania [6] both being conducted in a setting of mixed urbanicity (mix of rural and peri-urban). The study conducted in Benin was conducted in a rural setting [20]. Transmission intensity of malaria followed the rainy season in each location, which ranged from April-July and October-November (Benin) [20], May-October (Burkina-Faso) [21] and October-July (Tanzania) [6]. The species of parasite for each trial was *Plasmodium falciparum*, and every setting was considered to have a high level of transmission (*P. falciparum* prevalence of $> = 35\%$) according to the schema in the WHO: *a framework for malaria elimination* [22]. The main vectors of interest for the trials of Accrombessi, Cook [20] and Tiono, Ouédraogo [21] included both *Anopheles coluzzi and An. gambiae sensu stricto*. While the main vector considered by Mosha, Kulkarni [6] was *An. funestus*.

**Interventions and comparisons.** All studies implemented the intervention at the household level (e.g. distributed nets according to number of people residing in each household), and every study reported to have achieved a high level of coverage [23]. Accrombessi, Cook [20] assessed coverage as household access, and reported that one net was provided per every two people. Mosha, Kulkarni [6] and Tiono, Ouédraogo [21] assessed coverage as population access and reported a baseline intervention coverage of 62.2% and 95%, respectively.

Accrombessi, Cook [20] explored two interventions against a common comparator. The first intervention was the chlorfenapyr-pyrethroid ITN "Interceptor G2®". This ITN was made of polyester netting (100 deniers) impregnated with a wash-resistant formulation of 200 mg/m2 chlorfenapyr (a pyrole) and 100 mg/m2 alpha-cypermethrin (a pyrethroid). The second intervention was the pyriproxyfen-pyrethroid ITN "Royal Guard®". This ITN was made of polyethylene (120 deniers) incorporating 225 mg/m2 pyriproxyfen (an insect growth regulator) and 261 mg/m2 alpha-cypermethrin. Both interventions were compared against a control pyrethroid-only ITN treated with alpha-cypermethrin at a target dose of 200 mg/m2 of polyester fabric (100 deniers).

Mosha, Kulkarni [6] also investigated two interventions. The first intervention was the "Interceptor G2®" (same specifications as above) and the second was the "Royal Guard®" (same specifications as above). These interventions were compared to the "Interceptor®" (same specifications as above) and were also compared to "Olyset Plus®", a pyrethroid-PBO ITN (10g/kg of PBO and 20g/kg of permethrin incorporated into polyethylene fibres).

Finally, Tiono, Ouédraogo [21] evaluated the effectiveness of the pyriproxyfen-pyrethroid ITN "Olyset Duo®". These were polyethylene nets treated with a combination of 2% w/w permethrin and 1% w/w pyriproxyfen incorporated into polyethylene fibres. These were compared against pyrethroid-only ITNs "Olyset®" (2% w/w permethrin incorporated into polyethylene fibres). Tiono, Ouédraogo [21] employed a stepped-wedge design, where five clusters were randomised to the standard "Olyset®" ITNs at baseline and replaced with the "Olyset Duo®" ITNs by the end of the trial (June 2014 to December 2015). It is worth noting, that the Sumitomo Olyset Duo pyriproxyfen ITN has been withdrawn from the market and is not a WHO pre-qualified net.

**Outcomes.** The main outcome measured across all three studies was malaria case incidence. In the Accrombessi, Cook [20] trial, malaria case incidence was measured in a cohort of

30 children per cluster (aged 6 months to 10 years) that were randomly selected and actively followed up for 20 months. Similarly, Mosha, Kulkarni [6] measured malaria case incidence by actively following one child per household (aged 6-months to 14-years), from 35 randomly selected households per cluster, for up to 1-year. A second independent cohort of children from 40 randomly selected households per cluster were actively followed for 1-year, 1-year post intervention (e.g. from 1-year post to 2-years post). Tiono, Ouédraogo [21] however, measured malaria case incidence in approximately 2157 (balanced between groups) children aged six months to five years through passive case detection (presentation to health facility with malaria symptoms).

The other outcomes measured across all three studies were parasite prevalence and prevalence of anaemia. These outcomes were collected using cross-sectional surveys. Accrombessi, Cook [20] conducted a survey at 6-months and 18-months post implementation of the intervention. This survey included 70 people (of any age) randomly selected in each cluster. Mosha, Kulkarni [6] conducted cross-sectional surveys of up to two children per household if they were aged between 6 months and 14 years. These surveys were conducted at 12-months, 18-months and 24-months post implementation of the intervention. Mosha, Kulkarni [6] also collected data regarding all-cause mortality and malaria mortality during these surveys. Finally, Tiono, Ouédraogo [21] conducted four cross-sectional surveys of all children in the study area. These surveys were performed in June 2014, December 2014, May 2015 and July 2015 (time post intervention ranged from 5-weeks to 9-months). Due to the stepped-wedged nature of this trial, the data from May 2015 represents the survey in which 50% of the clusters randomised had received the intervention and 50% were still using the control ITN. Tiono, Ouédraogo [21] also collected all-cause mortality data during these surveys.

Malaria infection incidence and incidence of severe disease were outcomes stipulated in the protocol. However, these outcomes could not be synthesised as they were not reported by any of the included studies. Data regarding adverse events was also reported in two studies [6, 21] and contextual information regarding net quality was only reported in one [6]. Summary characteristics of the included studies has been provided in Table 6. The full details of these studies have been included in the characteristics of included studies tables (S3 in S1 File).

**Assessment of the risk of bias.** *Bias arising from the randomisation process.* Randomisation and allocation concealment were achieved through employing an independent statistician in Mosha, Kulkarni [6] and were judged as having low risk of bias for this domain. Accrombessi, Cook [20] stated in their protocol that "Restricted randomisation will be used..." but did not provide the review team with additional information regarding this procedure, or baseline demographics outside of children sex ratios for meta-determination of the randomisation sequence followed. Likewise, Tiono, Ouédraogo [21] achieved randomisation using "Stata version 10", however, no further details were provided regarding this process for whether allocation concealment took place. As such, both studies were judged as having 'some concerns' for this domain. Mosha, Kulkarni [6] also provided data for some of their outcomes that had not considered the intra-class correlation coefficient (ICC). This is particularly relevant for any comparison provided in this review against pyrethroid-PBO ITNs. As this raw data has not been appropriately controlled for the ICC, we have decided to consider a high risk of bias for this domain, wherever outcome data was relevant to the pyrethroid-PBO ITNs. (Fig 2).

*Bias arising from the timing of identification or recruitment of participants.* All studies were regarded as having low risk of bias for this domain. All studies identified clusters before the randomisation process and the baseline demographic data provided by Mosha, Kulkarni [6] and Tiono, Ouédraogo [21] suggest that there were no imbalances between groups which may suggest differential recruitment between groups (this data was not provided for Accrombessi, Cook [20]).

**Table 6. Summary characteristics of included studies.**

| Study (location) | Year(s) of study | Dual active ingredient insecticide treated nets | | Outcomes reported |
|---|---|---|---|---|
| | | DAI ITN characteristics | Number of clusters, population details and coverage | |
| Accrombessi 2023 (Benin, Cove, Zagnanado, and Ouinhi Districts) | 2020–2022 | 1. Interceptor G2 (200 mg/m2 chlorfenapyr and 100 mg/m2 alpha-cypermethrin) 2. Royal Guard (225 mg/m2 pyriproxyfen and 261 mg/m2 alpha-cypermethrin) | • Clusters = 60 (approximately 200 households per cluster) • Population = Approximately 1200 per cluster (actual numbers not provided) • Overall coverage = one LLIN per every two people (complete details not provided) | • Malaria case incidence rate • Parasite Prevalence • Prevalence of anaemia |
| Mosha 2022 (Tanzania, Misungwi district of Mwanza) | 2018–2020 | 1. Interceptor G2 (200 mg/m2 chlorfenapyr and 100 mg/m2 alpha-cypermethrin) 2. Royal Guard (225 mg/m2 pyriproxyfen and 261 mg/m2 alpha-cypermethrin) | • Clusters = 84 (119 households) • Population = 236,496 • Overall coverage = Coverage at baseline measured as 62.2% (Population access) | • Parasite prevalence (defined in the study as malaria prevalence) • Malaria case incidence • All-cause mortality • Malaria mortality • Prevalence of anaemia |
| Tiono 2018 (Burkina Faso, Cascades Region) | 2014–2017 | 1. Olyset Duo (2% w/w permethrin and 1% w/w pyriproxyfen) | • Clusters = 40 (consisting of 1–4 neighboring villages, aka compound). • Population = Population numbers not provided at time of randomization. 6062 households participated • Overall coverage = Coverage at baseline measured as 95% (Population access) | • Malaria case incidence rate • Parasite prevalence • All-cause mortality • Prevalence of anaemia |

*Bias arising from deviations from intended interventions.* All studies attempted to blind participants and staff to the intervention being received. Mosha, Kulkarni [6] utilised ITNs that were similar in appearance apart from a colour-coded loop. Tiono, Ouédraogo [21] stated that all ITNs were of similar shape, size and colour. While the methods of blinding for the Accrombessi, Cook [20] study have not been provided by the authors, the protocol for this trial states "Study participants will be blinded to the type of nets they have received. All field staff will be blinded to the allocation and analyses will be conducted on blinded data" [15]. As such, the risk of bias for all studies for this domain was low.

*Bias arising from missing outcome data.* In the Accrombessi, Cook [15] trial, malaria case incidence was measured in a cohort of 30 children per cluster that were randomly selected and followed up for 20 months. Parasite prevalence and prevalence of anaemia were measured following cross-sectional surveys of approximately 70 people (per cluster).

Mosha, Kulkarni [6] measured malaria case incidence by actively following one child, from 35 randomly selected households per cluster, for up to 1-year. A second independent cohort of children from 40 randomly selected households per cluster were actively followed for 1-year, 1-year post intervention (e.g. from 1-year post to 2-years post). The authors also collected parasite prevalence, prevalence of anaemia data and mortality data (all-cause and due to malaria) from cross-sectional surveys of up to two children per household if they were aged between 6 months and 14 years.

Finally, Tiono, Ouédraogo [21] measured malaria case incidence of children aged six months to five years through passive case detection (presentation to health facility with malaria

symptoms). A cross-sectional survey of all children in the study area was conducted (when the stepped-wedge design achieved 50:50 split between intervention arms), this survey collected data of parasite prevalence, prevalence of anaemia and mortality.

Across all studies and for all outcomes, data was not made available for every participant that belonged to a randomised cluster. However, the process of randomisation (that was evidenced in each study) and randomly selecting participants to provide outcome data, suggests that these results were not biased. Additionally, data was made available from every cluster, for all three studies. As such, all three studies have been judged to have a low risk of bias for this domain, and for every outcome reported.

*Bias arising from measurement of the outcome*. Measurement of all outcomes examined across every study were deemed to be appropriate (see details regarding outcomes above), and all studies employed an appropriate blinding method (see above) that suggests that blinding of the outcome assessor was likely. Therefore, all studies have been judged to have a low risk of bias for this domain, and for every outcome reported.

*Bias arising from selection of the reported results*. Accrombessi, Cook [20] includes data from 1-year post, 2-year post and overall data for all three outcomes reported above. All this data has been used in the review analyses and has followed a pre-specified analysis plan established in the protocol. Both Mosha, Kulkarni [6] and Tiono, Ouédraogo [21] reported multiple analyses of the data for each outcome (time points analysis). However, all the results were reported in the manuscripts transparently and included in this review as appropriate. These analyses were also conducted following the pre-specified analysis plan established in the trial protocols and the risk of bias for all three studies for this domain for each outcome is low.

*Overall bias*. Overall, the risk of bias was low for all studies across all outcomes (except for outcomes related to pyrethroid-PBO ITNs). Judgments for each included study have been summarised in Fig 2, with support for every judgment have been provided in S3 in S1 File.

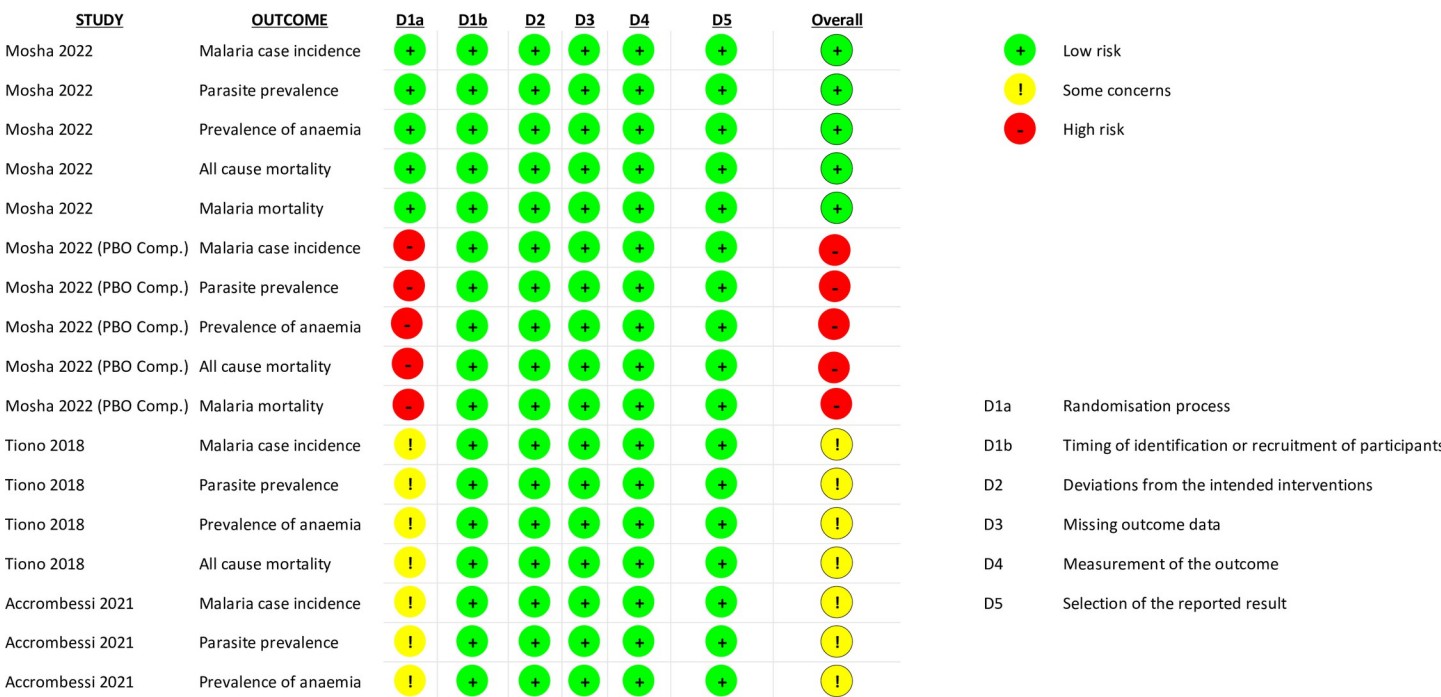

**Fig 2. Risk of bias judgements.** Summarised risk of bias judgements using the Cochrane RoB 2.0 tool for cluster randomised controlled trials. Provided for each study and each outcome where relevant.

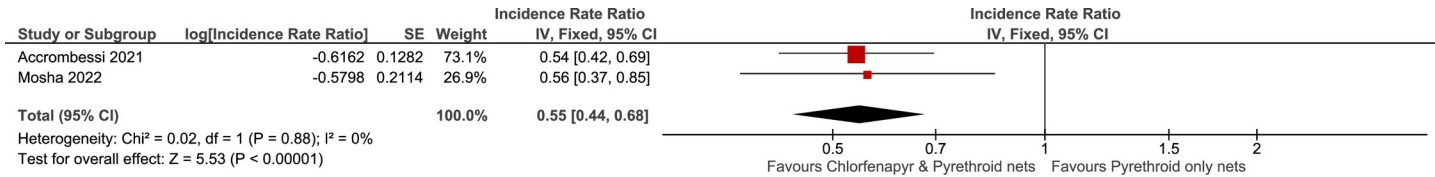

**Fig 3. Chlorfenapyr-pyrethroid ITNs versus pyrethroid-only ITNs: Malaria case incidence (overall).**

## Data synthesis and meta-analysis

**Comparison 1—Chlorfenapyr-pyrethroid ITNs versus pyrethroid-only ITNs.** *Malaria case incidence (overall)*. Two studies [6, 20] contributed data for this comparison and the below outcomes. There was a 45% reduction in malaria case incidence (overall) in clusters that received chlorfenapyr-pyrethroid ITNs compared to those that received pyrethroid-only ITNs (IRR = 0.55, 95%CI: 0.44–0.68, p <0.001, Fig 3). There was no important heterogeneity between these data ($I^2$ = 0%, $Chi^2$ = 0.01, p = 0.88) Subgroup analyses were conducted for vector species and setting. The ICEMAN credibility assessment identified these subgroups as both having very low credibility. This assessment suggested that effect modification is very unlikely and for the overall estimate to be used. Forest-plots for outcomes that have contributed to the evidence profiles (Tables 1–4) are presented below, all other forest-plots (including all subgroups) are provided in S4 in S1 File. The ICEMAN credibility assessments have been presented in S5 in S1 File.

*Malaria case incidence (1-year post intervention)*. There was a 53% reduction in malaria case incidence at 1-year post intervention in clusters that received chlorfenapyr-pyrethroid ITNs compared to those that received pyrethroid-only ITNs (IRR = 0.47, 95%CI: 0.35–0.63, p <0.001, Fig 4). There was no important heterogeneity between these data ($I^2$ = 0%, $Chi^2$ = 0.00, p = 0.94). Subgroup for vector species and setting using ICEMAN credibility assessment identified these subgroups as both having very low credibility suggesting very unlikely effect modification and for the overall estimate to be used.

*Malaria case incidence (2-years post intervention)*. There was a 33% reduction in malaria case incidence at 2-years post intervention in clusters that received chlorfenapyr-pyrethroid ITNs compared to those that received pyrethroid-only ITNs (IRR = 0.67, 95%CI: 0.61–0.75, p <0.001, Fig 5). There may be substantial heterogeneity between these data ($I^2$ = 75%, $Chi^2$ = 3.99, p = 0.05). Subgroup analyses were conducted for vector species and setting. The ICEMAN credibility assessment identified these subgroups as both having very low credibility. This suggested that effect modification is very unlikely and for the overall estimate to be used.

*Parasite prevalence (6-months follow-up)*. Accrombessi, Cook [20] reported a 53% reduction in parasite prevalence at 6-months follow-up, in clusters that received chlorfenapyr-pyrethroid ITNs compared to those that received pyrethroid-only ITNs (OR = 0.47, 95%CI: 0.32–0.69, p = 0.001, Fig 6).

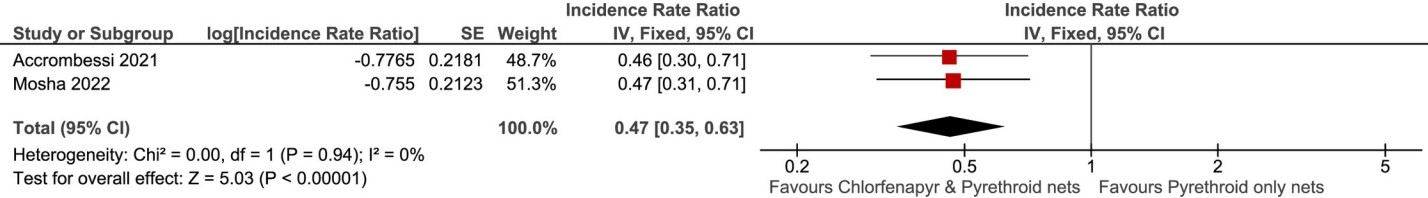

**Fig 4. Chlorfenapyr-pyrethroid ITNs versus pyrethroid-only ITNs: Malaria case incidence (1-year post).**

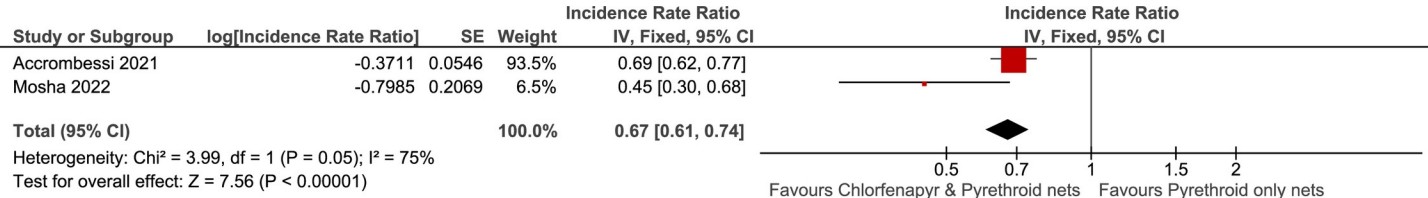

**Fig 5. Chlorfenapyr-pyrethroid ITNs versus pyrethroid-only ITNs: Malaria case incidence (2-year post).**

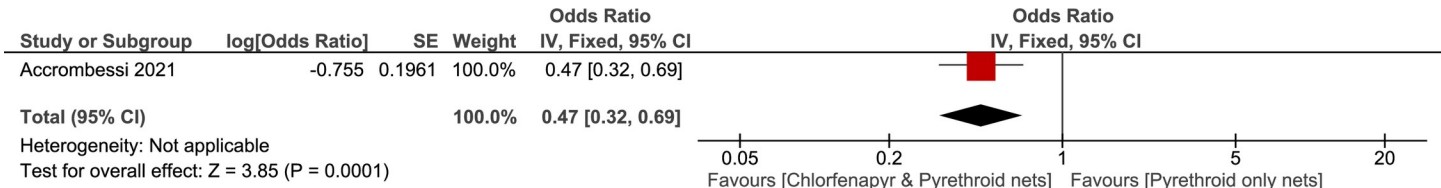

**Fig 6. Chlorfenapyr-pyrethroid ITNs versus pyrethroid-only ITNs: Parasite prevalence (6-months).**

*Parasite prevalence (12-months follow-up).* Mosha, Kulkarni [6] reported a 53% reduction in parasite prevalence at 12-months follow-up, in clusters that received chlorfenapyr-pyrethroid ITNs compared to those that received pyrethroid-only ITNs (OR = 0.47, 95%CI: 0.31–0.72, p = 0.004, Fig 7).

*Parasite prevalence (18-months follow-up).* There was a 37% reduction in parasite prevalence at 18-months follow-up, in clusters that received chlorfenapyr-pyrethroid ITNs compared to those that received pyrethroid-only ITNs (OR = 0.63, 95%CI: 0.49–0.80, p = 0.002, Fig 8). There was no important heterogeneity between these data ($I^2 = 0\%$, $Chi^2 = 0.14$, p = 0.71). Subgroup analyses were conducted for vector species and setting. The ICEMAN credibility assessment identified these subgroups as both having very low credibility. This suggested that effect modification is very unlikely and for the overall estimate to be used.

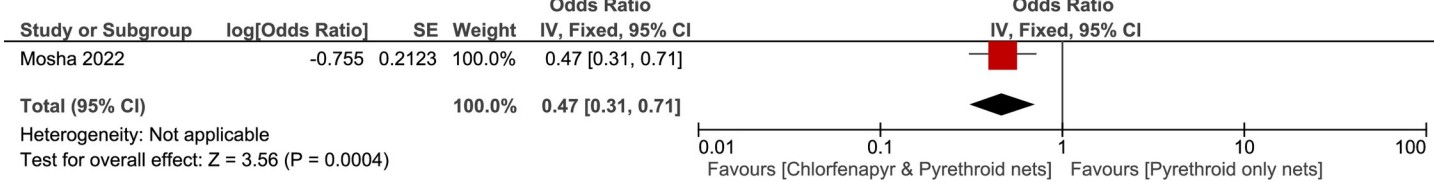

**Fig 7. Chlorfenapyr-pyrethroid ITNs versus pyrethroid-only ITNs: Parasite prevalence (12-months).**

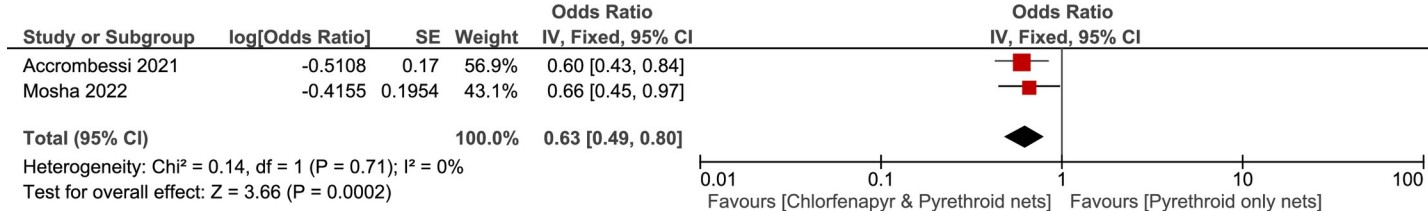

**Fig 8. Chlorfenapyr-pyrethroid ITNs versus pyrethroid-only ITNs: Parasite prevalence (18-months).**

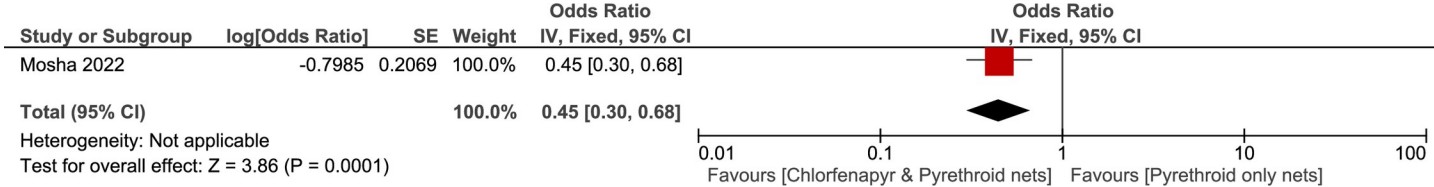

**Fig 9. Chlorfenapyr-pyrethroid ITNs versus pyrethroid-only ITNs: Parasite prevalence (24-months).**

*Parasite prevalence (24-months follow-up).* Mosha, Kulkarni [6] reported a 55% reduction in parasite prevalence at 24-months follow-up, in clusters that received chlorfenapyr-pyrethroid ITNs compared to those that received pyrethroid-only ITNs (OR = 0.45, 95%CI: 0.30–0.68, p = 0.001, Fig 9).

*Parasite prevalence (furthest possible follow-up).* There was a 47% reduction in parasite prevalence at the furthest possible follow-up time point, in clusters that received chlorfenapyr-pyrethroid ITNs compared to those that received pyrethroid-only ITNs (OR = 0.53, 95%CI: 0.41–0.69, p <0.001). There was no important heterogeneity between these data ($I^2$ = 13%, $Chi^2$ = 1.15, p = 0.28). Subgroup analyses were conducted for vector species and setting. The ICEMAN credibility assessment identified these subgroups as both having low credibility. This suggested that effect modification is unlikely and for the overall estimate to be used.

*Prevalence of anaemia (6-months follow-up).* Accrombessi, Cook [20] reported a 29% reduction in prevalence of anaemia at 6-months follow-up, in clusters that received chlorfenapyr-pyrethroid ITNs compared to those that received pyrethroid-only ITNs (OR = 0.71, 95%CI: 0.40–1.26, p = 0.16).

*Prevalence of anaemia (12-months follow-up).* Mosha, Kulkarni [6] reported a 45% increase in prevalence of anaemia at 12-months follow-up, in clusters that received chlorfenapyr-pyrethroid ITNs compared to those that received pyrethroid-only ITNs (OR = 1.55, 95%CI: 0.58–4.14, p = 0.38).

*Prevalence of anaemia (18-months follow-up).* There was a 17% reduction in prevalence of anaemia at 18-months follow-up, in clusters that received chlorfenapyr-pyrethroid ITNs compared to those that received pyrethroid-only ITNs (OR = 0.83, 95%CI: 0.53–1.28, p = 0.39). There was no heterogeneity between these data ($I^2$ = 0%, $Chi^2$ = 0.74, p = 0.39). Subgroup analyses were conducted for vector species and setting. The ICEMAN credibility assessment identified these subgroups as both having very low credibility. This suggested that effect modification is very unlikely and for the overall estimate to be used.

*Prevalence of anaemia (24-months follow-up).* Mosha, Kulkarni [6] reported a 6% reduction in prevalence of anaemia at 24-months follow-up, in clusters that received chlorfenapyr-pyrethroid ITNs compared to those that received pyrethroid-only ITNs (OR = 0.94, 95%CI: 0.52–1.70, p = 0.84).

*Prevalence of anaemia (furthest possible follow-up).* As this data has come from cross-sectional surveys, the survey from each study taken from the longest time post-intervention (where appropriate) was used (Mosha, Kulkarni [6]– 24 months, Accrombessi, Cook [20]– 18 months). There was a 1% reduction in prevalence of anaemia at the furthest possible follow-up time point, in clusters that received chlorfenapyr-pyrethroid ITNs compared to those that received pyrethroid-only ITNs (OR = 0.99, 95%CI: 0.62–1.58, p = 0.97). There was no important heterogeneity between these data ($I^2$ = 0%, $Chi^2$ = 0.08, p = 0.78). Subgroup analyses were conducted for vector species and setting. The ICEMAN credibility assessment identified these subgroups as both having very low credibility. This suggested that effect modification is very unlikely and for the overall estimate to be used.

**Comparison 2—Chlorfenapyr-pyrethroid ITNs versus pyrethroid-PBO ITNs.** *Malaria case incidence (overall)*. Only one study [6] contributed data for the outcomes under this comparison. Mosha, Kulkarni [6] reported a 32% reduction in malaria case incidence (overall), in clusters that received chlorfenapyr-pyrethroid ITNs compared to those that received pyrethroid-PBO ITNs (IRR = 0.68, 95%CI: 0.59–0.79, p <0.001, Fig 10).

*Malaria case incidence (1-year post intervention)*. Mosha, Kulkarni [6] reported a 2% reduction in malaria case incidence at 1-year post intervention, in clusters that received chlorfenapyr-pyrethroid ITNs compared to those that received pyrethroid-PBO ITNs (IRR = 0.98, 95% CI: 0.71–1.36, p = 0.90, Fig 11).

*Malaria case incidence (2-years post intervention)*. Mosha, Kulkarni [6] reported a 35% reduction in malaria case incidence at 2-years post intervention, in clusters that received chlorfenapyr-pyrethroid ITNs compared to those that received pyrethroid-PBO ITNs (IRR = 0.65, 95%CI: 0.55–0.77, p <0.001, Fig 12).

*Parasite prevalence (12-months follow-up)*. Mosha, Kulkarni [6] reported a 22% reduction in parasite prevalence at 12-months follow-up, in clusters that received chlorfenapyr-pyrethroid ITNs compared to those that received pyrethroid-PBO ITNs (OR = 0.78, 95%CI: 0.62–0.97, p = 0.03, Fig 13).

*Parasite prevalence (18-months follow-up)*. Mosha, Kulkarni [6] reported a 9% reduction in parasite prevalence at 18-months follow-up, in clusters that received chlorfenapyr-pyrethroid ITNs compared to those that received pyrethroid-PBO ITNs (OR = 0.91, 95%CI: 0.77–1.04, p = 0.23, Fig 14).

*Parasite prevalence (24-months follow-up)*. Mosha, Kulkarni [6] reported a 50% reduction in parasite prevalence at 24-months follow-up, in clusters that received chlorfenapyr-pyrethroid ITNs compared to those that received pyrethroid-PBO ITNs (OR = 0.50, 95%CI: 0.42–0.60, p <0.001, Fig 15).

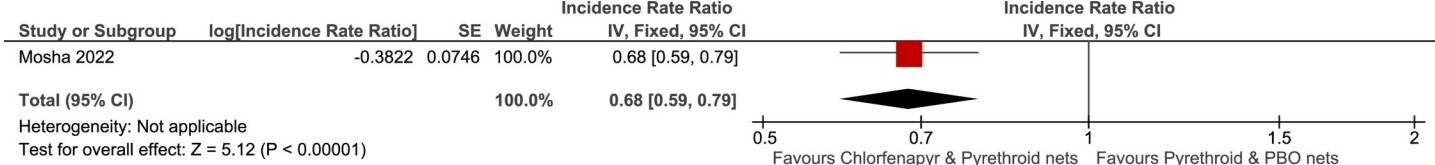

**Fig 10. Chlorfenapyr-pyrethroid ITNs versus pyrethroid-PBO ITNs: Malaria case incidence (overall).**

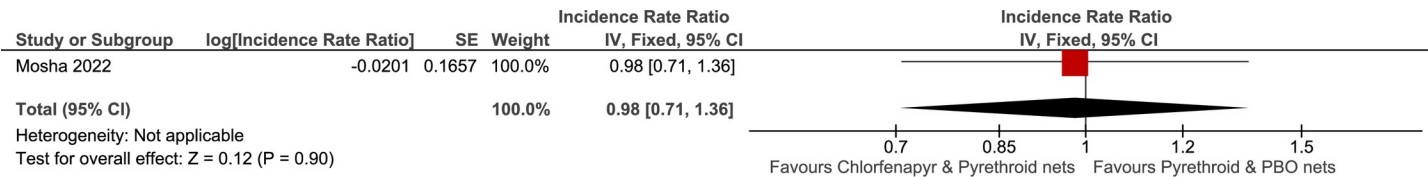

**Fig 11. Chlorfenapyr-pyrethroid ITNs versus pyrethroid-PBO ITNs: Malaria case incidence (1-year post).**

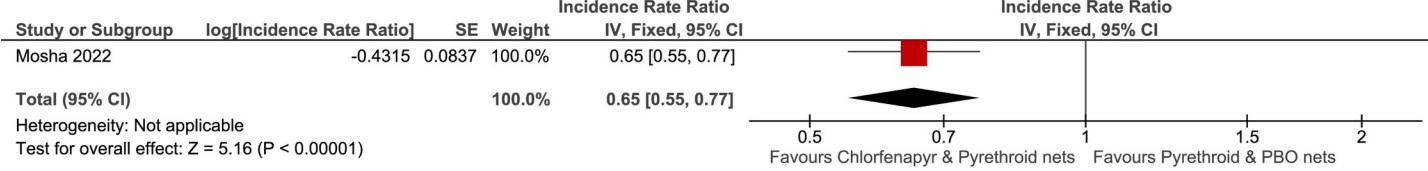

**Fig 12. Chlorfenapyr-pyrethroid ITNs versus pyrethroid-PBO ITNs: Malaria case incidence (2-year post).**

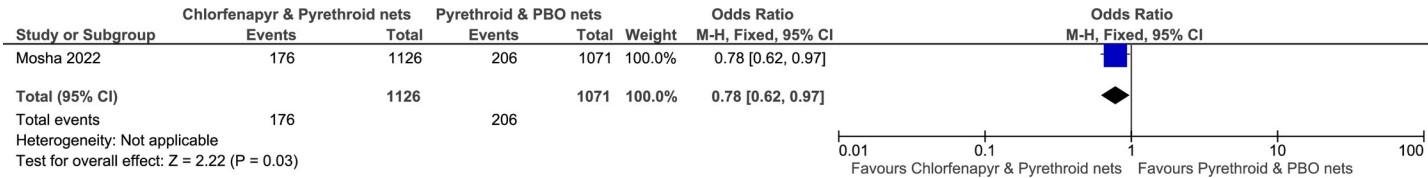

**Fig 13. Chlorfenapyr-pyrethroid ITNs versus pyrethroid-PBO ITNs: Parasite prevalence (12-months).**

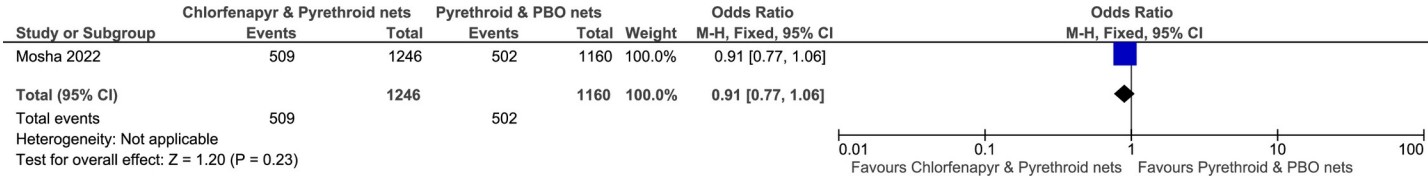

**Fig 14. Chlorfenapyr-pyrethroid ITNs versus pyrethroid-PBO ITNs: Parasite prevalence (18-months).**

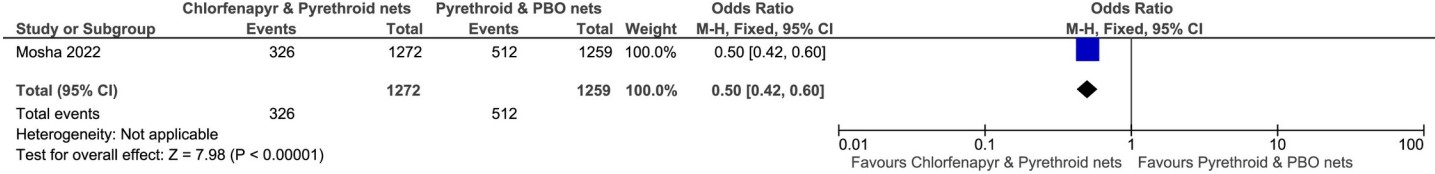

**Fig 15. Chlorfenapyr-pyrethroid ITNs versus pyrethroid-PBO ITNs: Parasite prevalence (24-months).**

*Prevalence of anaemia (12-months follow-up).* Mosha, Kulkarni [6] reported a 1% reduction in prevalence of anaemia at 12-months follow-up, in clusters that received chlorfenapyr-pyrethroid ITNs compared to those that received pyrethroid-PBO ITNs (OR = 0.99, 95%CI: 0.42–2.37, p = 0.03).

*Prevalence of anaemia (18-months follow-up).* Mosha, Kulkarni [6] reported a 30% increase in prevalence of anaemia at 18-months follow-up, in clusters that received chlorfenapyr-pyrethroid ITNs compared to those that received pyrethroid-PBO ITNs (OR = 1.30, 95%CI: 0.75–2.26, p = 0.35).

*Prevalence of anaemia (24-months follow-up).* Mosha, Kulkarni [6] reported a 4% increase in prevalence of anaemia at 24-months follow-up, in clusters that received chlorfenapyr-pyrethroid ITNs compared to those that received pyrethroid-PBO ITNs (OR = 1.04, 95%CI: 0.60–1.80, p = 0.89).

**Comparison 3—Pyriproxyfen-pyrethroid ITNs versus pyrethroid-only ITNs.** *Malaria case incidence (overall).* All three included studies contributed data to this outcome. There was a 10% reduction in malaria case incidence (overall) in clusters that received pyriproxyfen-pyrethroid ITNs compared to those that received pyrethroid-only ITNs (IRR = 0.90, 95%CI: 0.73–1.13, p = 0.37, Fig 16). There was no important heterogeneity between these data ($I^2$ = 0%, $Chi^2$ = 0.33, p = 0.85). The data has been separated into subgroups based on active-ingredient composition and manufacturer. However, ICEMAN credibility assessments determined very low credibility, suggesting that that there was very likely no effect modification between these subgroups and the overall effect should be used. Subgroup analysis was also conducted for vector species and setting; however, ICEMAN credibility assessments determined these subgroups to also be very-low. As such, it is very likely that no effect modification was present.

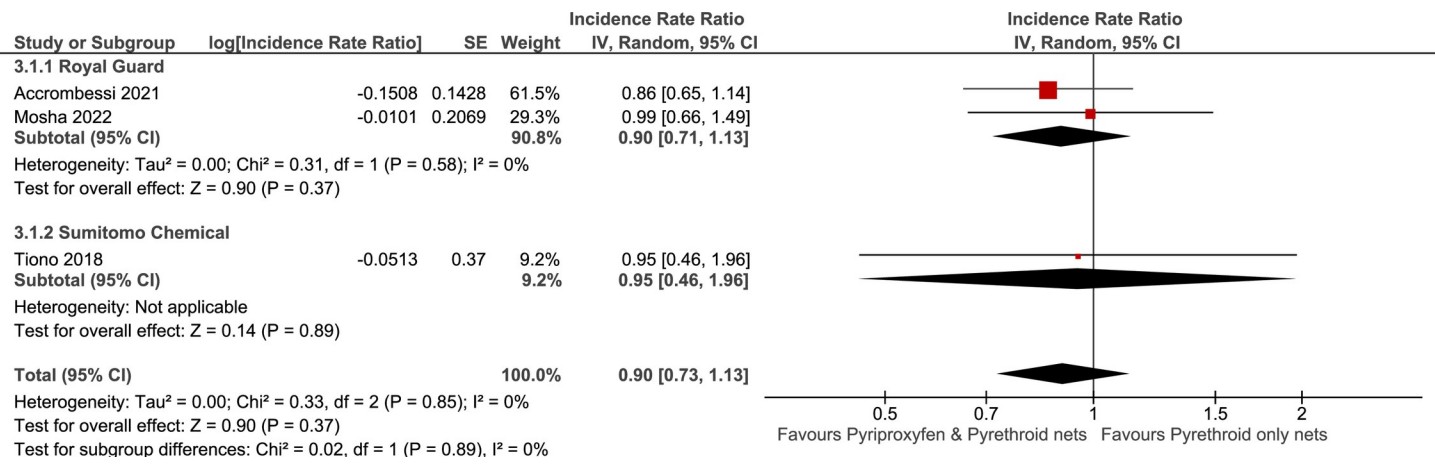

**Fig 16. Pyriproxyfen-pyrethroid ITNs versus pyrethroid-only ITNs: Malaria case incidence (overall).**

*Malaria case incidence (1-year post intervention).* Only data from Mosha, Kulkarni [6] and Accrombessi, Cook [20] have provided data for outcomes regarding time from implementation of the intervention. There was a 34% reduction in malaria case incidence at 1-year post intervention in clusters that received pyriproxyfen-pyrethroid ITNs compared to those that received pyrethroid-only ITNs (IRR = 0.66, 95%CI: 0.47–0.85, p = 0.02, Fig 17). There may be some moderate heterogeneity between these data ($I^2$ = 40%, $Chi^2$ = 1.67, p = 0.2). Subgroup analyses were conducted for vector species and setting. The ICEMAN credibility assessment identified these subgroups as both having very low credibility. This suggested that effect modification is very unlikely and for the overall estimate to be used.

*Malaria case incidence (2-years post intervention).* There was a 6% reduction in malaria case incidence at 2-years post intervention in clusters that received pyriproxyfen-pyrethroid ITNs compared to those that received pyrethroid-only ITNs (IRR = 0.94, 95%CI: 0.75–1.17, p = 0.57, Fig 18). There was no important heterogeneity between these data ($I^2$ = 0%, $Chi^2$ = 0.86, p = 0.35). Subgroup analyses were conducted for vector species and setting. The ICEMAN credibility assessment identified these subgroups as both having very low credibility. This suggested that effect modification is very unlikely and for the overall estimate to be used.

*Parasite prevalence at 6-months follow-up.* Accrombessi, Cook [20] reported an 8% reduction in parasite prevalence at 6-months follow-up, in clusters that received pyriproxyfen-pyrethroid ITNs compared to those that received pyrethroid-only ITNs (OR = 0.92, 95%CI: 0.63–1.34, p = 0.67, Fig 19).

*Parasite prevalence (12-months follow-up).* Mosha, Kulkarni [6] reported a 31% reduction in parasite prevalence at 12-months follow-up, in clusters that received pyriproxyfen-pyrethroid ITNs compared to those that received pyrethroid-only ITNs (OR = 0.69, 95%CI: 0.46–1.04, p = 0.08, Fig 20).

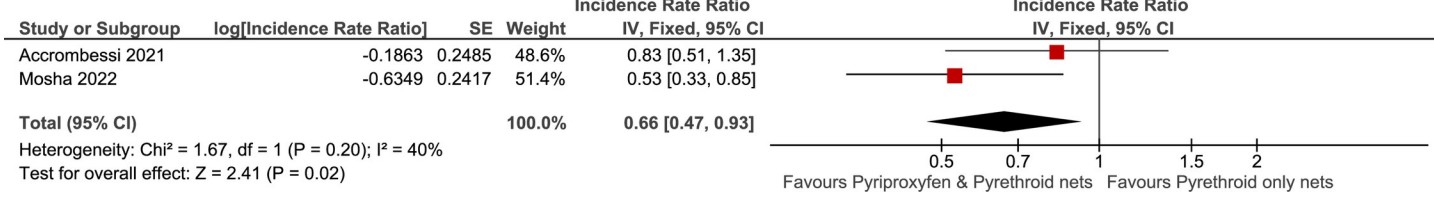

**Fig 17. Pyriproxyfen-pyrethroid ITNs versus pyrethroid-only ITNs: Malaria case incidence (1-year post).**

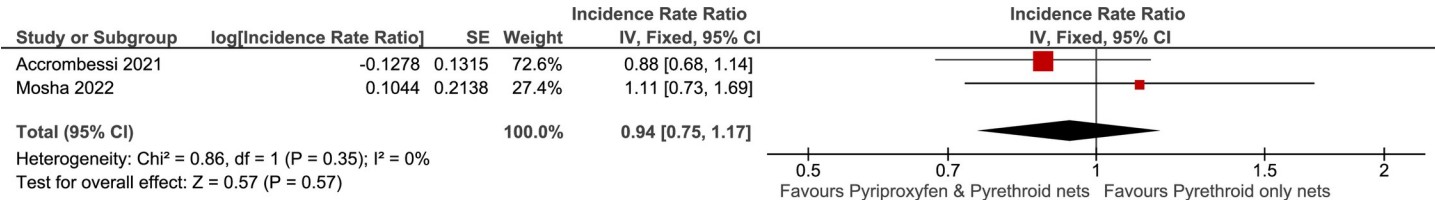

**Fig 18. Pyriproxyfen-pyrethroid ITNs versus pyrethroid-only ITNs: Malaria case incidence (2-year post).**

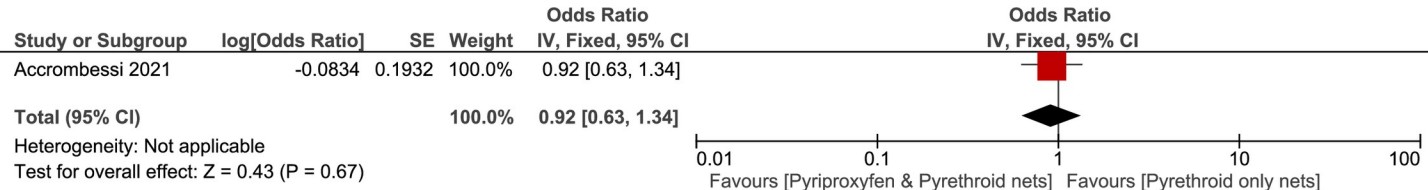

**Fig 19. Pyriproxyfen-pyrethroid ITNs versus pyrethroid-only ITNs: Parasite prevalence (6-months).**

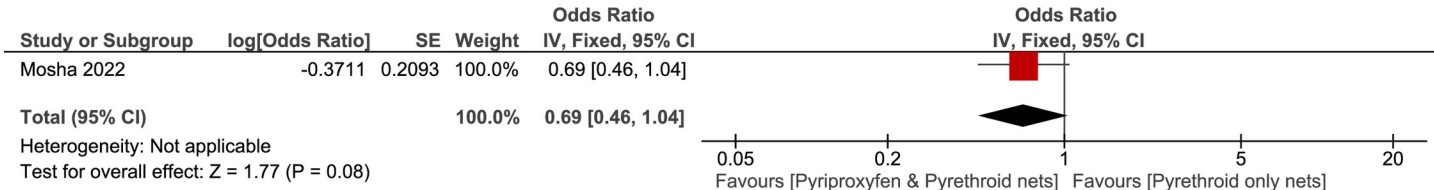

**Fig 20. Pyriproxyfen-pyrethroid ITNs versus pyrethroid-only ITNs: Parasite prevalence (12-months).**

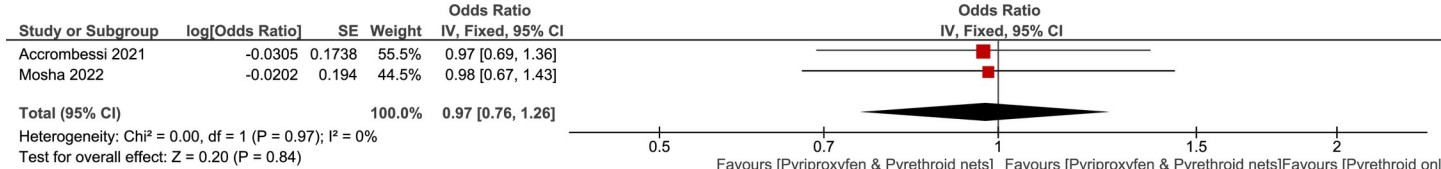

**Fig 21. Pyriproxyfen-pyrethroid ITNs versus pyrethroid-only ITNs: Parasite prevalence (18-months).**

*Parasite prevalence (18-months follow-up).* There was a 3% reduction in parasite prevalence at 18-months follow-up, in clusters that received pyriproxyfen-pyrethroid ITNs compared to those that received pyrethroid-only ITNs (OR = 0.97, 95%CI: 0.76–1.26, p = 0.84, Fig 21). There was no important heterogeneity between these data ($I^2$ = 0%, $Chi^2$ = 0.0, p = 0.97). Sub-group analyses were conducted for vector species and setting. The ICEMAN credibility assessment identified these subgroups as both having very low credibility. This suggested that effect modification is very unlikely and for the overall estimate to be used.

*Parasite prevalence (24-months follow-up).* Mosha, Kulkarni [6] reported an 21% reduction in parasite prevalence at 24-months follow-up, in clusters that received pyriproxyfen-pyrethroid ITNs compared to those that received pyrethroid-only ITNs (OR = 0.79, 95%CI: 0.54–1.16, p = 0.22, Fig 22).

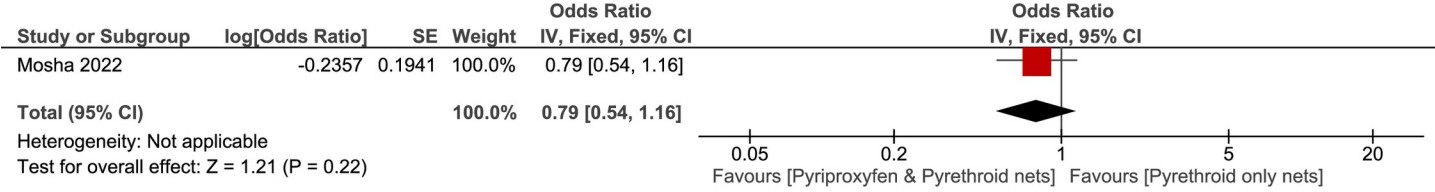

| Study or Subgroup | log[Odds Ratio] | SE | Weight | Odds Ratio IV, Fixed, 95% CI | Odds Ratio IV, Fixed, 95% CI |
|---|---|---|---|---|---|
| Mosha 2022 | -0.2357 | 0.1941 | 100.0% | 0.79 [0.54, 1.16] | |
| Total (95% CI) | | | 100.0% | 0.79 [0.54, 1.16] | |

Heterogeneity: Not applicable
Test for overall effect: Z = 1.21 (P = 0.22)

Favours [Pyriproxyfen & Pyrethroid nets]  Favours [Pyrethroid only nets]

**Fig 22. Pyriproxyfen-pyrethroid ITNs versus pyrethroid-only ITNs: Parasite prevalence (24-months).**

*Parasite prevalence (furthest possible follow-up)*. All three included studies contributed data to this outcome. The survey from each study taken from the longest time post-intervention (where appropriate) was used (Mosha, Kulkarni [6]– 24 months, Accrombessi, Cook [20]– 18 months). However, due to the stepped-wedged nature of Tiono, Ouédraogo [21], the data from survey conducted in May 2015 has been used. This data represents the longest point in the trial from the implementation of the intervention in which 50% of the clusters randomised had received the intervention and 50% were still using the control ITN. There was a 11% reduction in parasite prevalence at the furthest possible follow-up time point, in clusters that received pyriproxyfen-pyrethroid ITNs compared to those that received pyrethroid-only ITNs (OR = 0.89, 95%CI: 0.79–1.01, p = 0.07). There was no important heterogeneity between these data ($I^2$ = 0%, $Chi^2$ = 0.63, p = 0.73) The data has been separated into subgroups based on active-ingredient composition and manufacturer. However, ICEMAN credibility assessments determined very low credibility, suggesting that that there was very likely no effect modification between these subgroups and the overall effect should be used. Subgroup analysis was also conducted for vector species and setting; however, ICEMAN credibility assessments determined these subgroups to be low and very-low respectively. As such, it is likely to very likely that no effect modification was present.

*Prevalence of anaemia at 6-months follow-up*. Accrombessi, Cook [20] reported a 24% increase in prevalence of anaemia at 6-months follow-up, in clusters that received pyriproxyfen-pyrethroid ITNs compared to those that received pyrethroid-only ITNs (OR = 1.24, 95% CI: 0.71–2.17, p = 0.45).

*Prevalence of anaemia (12-months follow-up)*. Mosha, Kulkarni [6] reported a 15% increase in prevalence of anaemia at 12-months follow-up, in clusters that received pyriproxyfen-pyrethroid ITNs compared to those that received pyrethroid-only ITNs (OR = 1.15, 95%CI: 0.40–3.31, p = 0.80).

*Prevalence of anaemia (18-months follow-up)*. There was a 13% reduction in prevalence of anaemia at 18-months follow-up, in clusters that received pyriproxyfen-pyrethroid ITNs compared to those that received pyrethroid-only ITNs (OR = 0.87, 95%CI: 0.56–1.33, p = 0.51). There was no important heterogeneity between these data ($I^2$ = 0%, $Chi^2$ = 0.01, p = 0.92). Subgroup analyses were conducted for vector species and setting. The ICEMAN credibility assessment identified these subgroups as both having very low credibility. This suggested that effect modification is very unlikely and for the overall estimate to be used.

*Prevalence of anaemia (24-months follow-up)*. Mosha, Kulkarni [6] reported a 15% increase in prevalence of anaemia at 24-months follow-up, in clusters that received pyriproxyfen-pyrethroid ITNs compared to those that received pyrethroid-only ITNs (OR = 1.15, 95%CI: 0.66–2.00, p = 0.62).

*Prevalence of anaemia (furthest possible follow-up)*. There was a 23% reduction in prevalence of anaemia at the furthest possible follow-up time point, in clusters that received pyriproxyfen-pyrethroid ITNs compared to those that received pyrethroid-only ITNs (OR = 0.77, 95%CI: 0.58–1.03, p = 0.08). There may have been moderate heterogeneity between these data

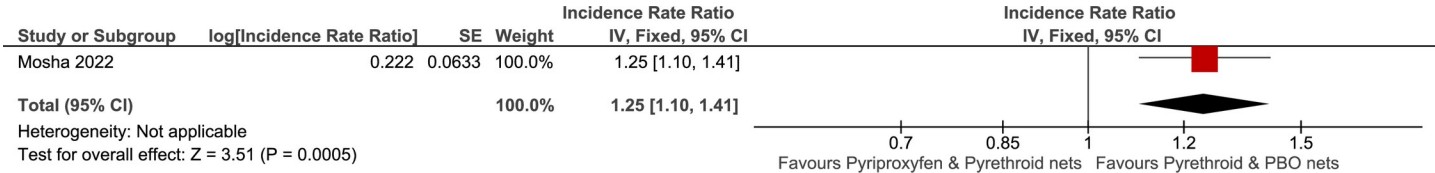

**Fig 23. Pyriproxyfen-pyrethroid ITNs versus pyrethroid-PBO ITNs: Malaria case incidence (overall).**

($I^2$ = 38%, Chi$^2$ = 3.21, p = 0.20). The data has been separated into subgroups based on active-ingredient composition and manufacturer. However, ICEMAN credibility assessments determined low credibility, suggesting that that there was likely no effect modification between these subgroups and the overall effect should be used. Subgroup analysis was also conducted for vector species and setting; however, ICEMAN credibility assessments determined these subgroups to be very-low. As such, it is very likely that no effect modification was present.

**Comparison 4—Pyriproxyfen-pyrethroid ITNs versus pyrethroid-PBO ITNs.** *Malaria case incidence (overall).* Only one study [6] directly compared any DAI ITN against an ITN that combined a pyrethroid and PBO in the one net. Mosha, Kulkarni [6] reported a 25% increase in malaria case incidence (overall) in clusters that received pyriproxyfen-pyrethroid ITNs compared to those that received pyrethroid-PBO ITNs (IRR = 1.25, 95%CI: 1.10–1.41, p = 0.0005, Fig 23).

*Malaria case incidence (1-year post intervention).* Mosha, Kulkarni [6] reported a 104% increase in malaria case incidence at 1-year post intervention, in clusters that received pyriproxyfen-pyrethroid ITNs compared to those that received pyrethroid-PBO ITNs (IRR = 2.04, 95%CI: 1.55–2.68, p <0.001, Fig 24).

*Malaria case incidence (2-years post intervention).* Mosha, Kulkarni [6] reported a 10% increase in malaria case incidence at 2-years post intervention, in clusters that received pyriproxyfen-pyrethroid ITNs compared to those that received pyrethroid-PBO ITNs (IRR = 1.10, 95%CI: 0.95–1.27, p = 0.19, Fig 25).

*Parasite prevalence (12-months follow-up).* Mosha, Kulkarni [6] reported a 16% increase in parasite prevalence at 12-months follow-up, in clusters that received pyriproxyfen-pyrethroid ITNs compared to those that received pyrethroid-PBO ITNs (OR = 1.16, 95%CI: 0.94–1.44, p = 0.16, Fig 26).

*Parasite prevalence (18-months follow-up).* Mosha, Kulkarni [6] reported a 34% increase in parasite prevalence at 18-months follow-up, in clusters that received pyriproxyfen-pyrethroid ITNs compared to those that received pyrethroid-PBO ITNs (OR = 1.34, 95%CI: 1.14–1.58, p = 0.0005, Fig 27).

*Parasite prevalence (24-months follow-up).* Mosha, Kulkarni [6] reported a 12% reduction in parasite prevalence at 24-months follow-up, in clusters that received pyriproxyfen-pyrethroid ITNs compared to those that received pyrethroid-PBO ITNs (OR = 0.88, 95%CI: 0.75–1.03, p = 0.11, Fig 28).

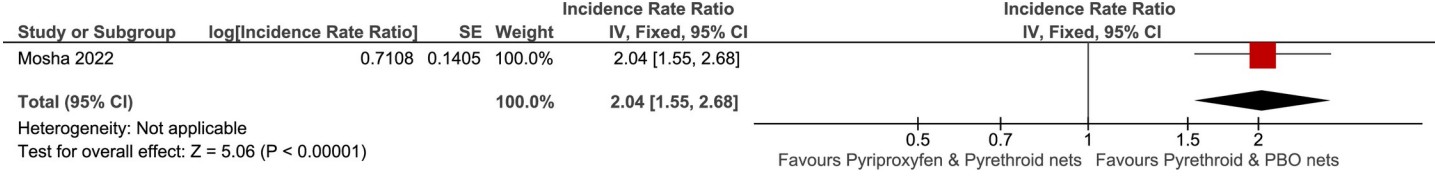

**Fig 24. Pyriproxyfen-pyrethroid ITNs versus pyrethroid-PBO ITNs: Malaria case incidence (1-year post).**

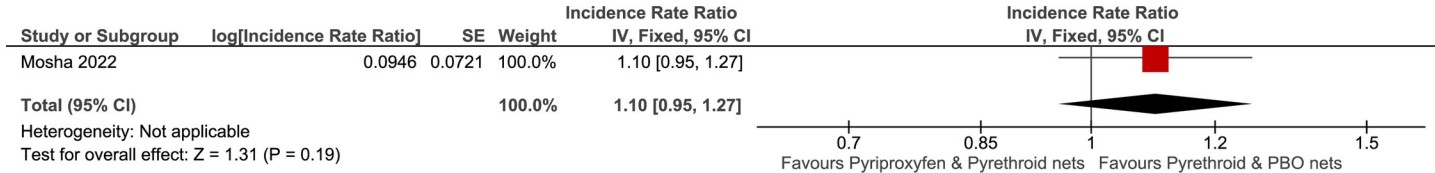

**Fig 25. Pyriproxyfen-pyrethroid ITNs versus pyrethroid-PBO ITNs: Malaria case incidence (2-year post).**

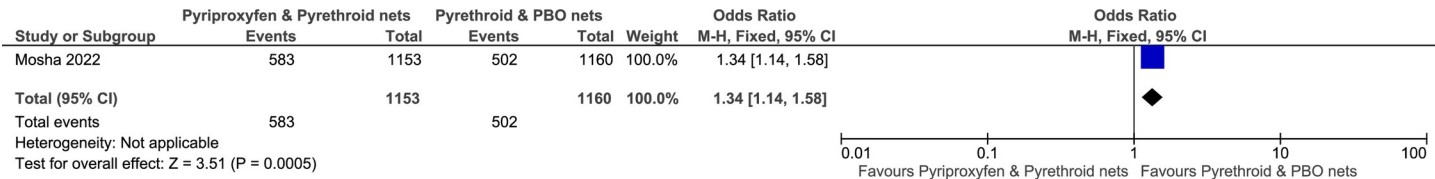

**Fig 26. Pyriproxyfen-pyrethroid ITNs versus pyrethroid-PBO ITNs: Parasite prevalence (12-months).**

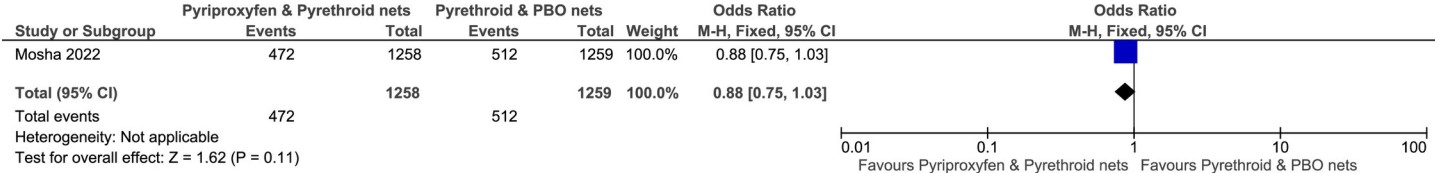

**Fig 27. Pyriproxyfen-pyrethroid ITNs versus pyrethroid-PBO ITNs: Parasite prevalence (18-months).**

**Fig 28. Pyriproxyfen-pyrethroid ITNs versus pyrethroid-PBO ITNs: Parasite prevalence (24-months).**

*Prevalence of anaemia (12-months follow-up).* Mosha, Kulkarni [6] reported a 20% reduction in prevalence of anaemia at 12-months follow-up, in clusters that received pyriproxyfen-pyrethroid ITNs compared to those that received pyrethroid-PBO ITNs (OR = 0.80, 95%CI: 0.31–2.04, p = 0.63).

*Prevalence of anaemia (18-months follow-up).* Mosha, Kulkarni [6] reported a 67% increase in prevalence of anaemia at 18-months follow-up, in clusters that received pyriproxyfen-pyrethroid ITNs compared to those that received pyrethroid-PBO ITNs (OR = 1.67, 95%CI: 0.97–2.87, p = 0.06).

*Prevalence of anaemia (24-months follow-up).* Mosha, Kulkarni [6] reported a 38% increase in prevalence of anaemia at 24-months follow-up, in clusters that received pyriproxyfen-pyrethroid ITNs compared to those that received pyrethroid-PBO ITNs (OR = 1.38, 95%CI: 0.82–2.32, p = 0.22).

**Mortality.** Both Mosha, Kulkarni [6] and Tiono, Ouédraogo [21] reported data regarding mortality outcomes. Mosha, Kulkarni [6] reported a total of five deaths among cohort children

during the study. While these deaths have been reported per group and year, the reasons (of death) have not been separated by group or year. As reported by the authors three deaths were from drowning, one was due to severe malaria, and one due to pneumonia, all of which were judged to be unrelated to the study interventions. Tiono, Ouédraogo [21] reported that there were 19 serious adverse events across all study participants (discussed below), and six of these resulted in deaths (n = 1 [Standard ITN], n = 5 [DAI ITN]). However, the months in which these deaths were recorded was not provided which prevented this data being presented as a forest-plot, as an appropriate denominator could not be determined, due to the stepped-wedge design of the trial.

**Adverse events.** Mosha, Kulkarni [6] reported that at 3 months post-intervention, adverse events were reported in 90 (44.1%) of those assigned the pyrethroid-only ITN, 80 (38.8%) of those assigned pyriproxyfen-pyrethroid ITNs, 17 (8.5%) assigned the chlorfenapyr-pyrethroid ITN, and 17 (8.5%) of those assigned the pyrethroid-PBO ITN. They also narratively reported that skin irritation was the most reported adverse event, however no adverse event was considered to be serious.

Tiono, Ouédraogo [21] reported 21 non-serious adverse events in the pyrethroid-only ITN group and one in the pyriproxyfen-pyrethroid ITN group. The adverse event in this group was a case of bronchitis. The adverse events in the pyrethroid-only ITN group included bronchitis, conjunctivitis, eye pruritus, pelvic pain, pruritus, rhinitis, cough and watering eyes all of which were resolved by study staff. Tiono, Ouédraogo [21] also reported 10 serious adverse events in the pyrethroid-only ITN group and 9 in the pyriproxyfen-pyrethroid ITN group. These included severe malaria with other comorbidities, uncomplicated malaria with vomiting, gastroenteritis with severe dehydration and pneumonia. However, these were not disaggregated between groups.

**Contextual factors.** Only Mosha, Kulkarni [6] reported data related to contextual factors of the ITNs used during the trial. The authors reported the proportion of ITNs that were torn (defined as hole area $\geq$790 cm$^2$). There were 86 (28%) torn in the pyrethroid-only ITN group, 109 (39%) were torn in the pyriproxyfen-pyrethroid ITN group, 96 (34%) were torn in the chlorfenapyr-pyrethroid ITN group, and 81 (43%) were torn in the pyrethroid-PBO group. No study reported any data regarding values or preferences regarding the interventions.

**Cost effectiveness.** Cost effectiveness was only reported by Mosha, Kulkarni [6] who modelled cost-effectiveness over the 2-year trial period. Malaria incidence estimates for each trial year were combined with probabilities of progression to severe disease and death that were collected from secondary sources. The authors used age-stratified malaria estimates for all countries from the "Global Burden of Disease Study 2019" incidence in people older than 10 years was estimated as a function of incidence in children aged 6 months to 10 years, and deaths in people older than 10 years were estimated as a fixed ratio to modelled deaths in children aged 6 months to 10 years. The authors then used Monte Carlo simulation to conduct probabilistic analyses, which reflected combined uncertainty in stochastic parameters. Analyses were re-run, varying one key parameter at a time, to examine the robustness of results to plausible variations in individual parameters. A threshold analysis identified the price of each net at which cost-effectiveness conclusions would change.

Mosha, Kulkarni [6] stated that chlorfenapyr-pyrethroid ITNs were estimated to avert the most DALYs (disability adjusted life years) (mean 152 DALYs averted [SD 72] per 10,000 total population). This was followed by pyrethroid-PBO ITNs (37 DALYs averted [72] per 10,000 population). Pyriproxyfen-pyrethroid ITNs incurred 9 more DALYs [71] per 10,000 population than compared to pyrethroid-only ITNs.

Mosha, Kulkarni [6] also reported that pyrethroid-only ITNs were the least costly to procure at $2.07 per net ($US), this was followed by pyrethroid-PBO ITNs at $2.98 per net.

Chlorfenapyr-pyrethroid ITNs were the next most expensive at $3.02 per net, while pyriproxy-fen-pyrethroid ITNs were the most expensive $3.68. However, when considering the costs of malaria diagnosis and prevention, and compared to pyrethroid-only ITNs over 2 years the chlorfenapyr-pyrethroid ITNs were the least costly (incremental cost $2894 [SD 1129] per 10 000 population). This was followed by pyrethroid-PBO ITNs ($4816 [SD 1360]) and pyriprox-yfen-pyrethroids ITNs were the most expensive ($9621 [SD 1327]).

Mosha, Kulkarni [6] conclude by stating that chlorfenapyr-pyrethroid ITNs were the more cost-effective strategy over a 2-year period. Chlorfenapyr-pyrethroid ITNs would cost an additional $19 (95%CI from $105 to $1) to public providers or $28 (95%CI from $11 to $120) to donors per DALY averted compared to pyrethroid-only ITNs. The pyrethroid-PBO ITNs were less effective and more costly and were estimated to cost an additional $130 (95%CI from $12 to -$59) to public providers and $136 to donors (95%CI from $22 to -$58) per DALY averted.

## Discussion

This is the first systematic review to assess the effectiveness of DAI ITNs against pyrethroid-only or pyrethroid-PBO ITNs. Three, cluster-randomised controlled trials were included in this review. Two studies employed a typical design and were conducted in Benin [20] and Tanzania [6]. One study utilised a stepped-wedge design and was conducted in Burkina Faso [21]. All studies were conducted in settings with high transmission of *P. falciparum*. The interventions investigated included chlorfenapyr-pyrethroid ITNs (Interceptor G2) and pyriproxyfen-pyrethroid ITNs (Royal Guard and/or Olyset Duo). These interventions have been compared against pyrethroid-only ITNs (Interceptor and/or Olyset) or pyrethroid-PBO ITNs (Olyset Plus). All studies utilised similar modes of intervention implementation and achieved a high coverage of the ITNs at baseline.

When collapsing the data across time points, the evidence suggests that clusters that receive a chlorfenapyr-pyrethroid ITN (Fig 3) will likely result in a reduction of malaria case incidence compared to clusters that receive a pyrethroid-only ITN. This finding was associated with high certainty of the evidence (Table 1). Compared to the control group, the reduction in malaria case incidence appears to be greater for the clusters receiving the chlorfenapyr-pyrethroid ITN (Fig 3) than the reduction observed for clusters receiving the pyriproxyfen-pyrethroid ITN (when collapsing across time points) (Fig 16). However, this difference as with all results discussed in this section, needs to be contextualised by a guideline panel.

At 1-year post intervention, the evidence suggests that clusters that receive either a chlorfenapyr-pyrethroid ITN (Fig 4) or pyriproxyfen-pyrethroid ITN (Fig 17) will result in a reduction of malaria case incidence compared to clusters that receive pyrethroid-only ITNs. Both findings were associated with high certainty of the evidence (Tables 1 and 3). At 2-years post-intervention the evidence suggests that, compared to clusters receiving pyrethroid-only ITNs, clusters that receive chlorfenapyr-pyrethroid ITNs (Fig 5) will also result in a reduction of malaria case incidence (Table 1). However, this was not observed in clusters that received pyriproxyfen-pyrethroid ITNs (Fig 18, Table 3). These findings were associated with High and Low certainty evidence respectively. For both time points it appears that the reduction in malaria case incidence is greater for the clusters receiving the DAI ITN containing chlorfenapyr and pyrethroid, than the reduction observed for clusters that received the DAI ITN containing pyriproxyfen and pyrethroid (both compared to clusters receiving pyrethroid-only ITNs).

Parasite prevalence at 6-months follow-up was only reported by Accrombessi, Cook [20] for both formulations of DAI ITNs. The authors have reported that both clusters receiving both formulations resulted in a reduction in parasite prevalence. However, chlorfenapyr-

pyrethroid ITNs appear to offer a greater reduction (Fig 6, Table 1) with higher certainty of the evidence, compared to pyriproxyfen-pyrethroid ITNs (Fig 19, Table 3) with moderate certainty of the evidence. Parasite prevalence at 12-months and 24-months follow-up was only reported by Mosha, Kulkarni [6] for both formulations of DAI ITNs. For the data at both the 12-month time point and 24-month time point, the authors have reported that clusters receiving chlorfenapyr-pyrethroid ITNs (Figs 7 and 9) resulted in a reduction of parasite prevalence, compared to clusters receiving the pyrethroid-only ITN. These findings were associated with high and moderate certainty evidence respectively (Table 1). For clusters that received pyriproxyfen-pyrethroid ITNs (Figs 20 and 22), there was no difference in parasite prevalence compared to control clusters at these time points. These findings were both associated with moderate certainty in the evidence (Table 3).

For parasite prevalence at 18-months, the evidence suggests that clusters that receive either formulation of DAI ITN will result in a reduction of parasite prevalence (Figs 8 and 21). However, chlorfenapyr-pyrethroid ITNs appear to offer a greater reduction in parasite prevalence, associated with higher certainty of the evidence Table 1), compared to the reduction offered by pyriproxyfen-pyrethroid ITNs (Table 3), which was only associated with low evidence.

Only one study [6] compared both formulations of DAI ITN against pyrethroid-PBO ITNs. Compared to pyrethroid-PBO ITNs, the authors have reported that clusters that received chlorfenapyr-pyrethroid ITNs were associated with reduced malaria case incidence when collapsed across time points (Fig 10), for 1-year post intervention (Fig 11), and for 2-years post intervention (Fig 12). These findings were associated with moderate, very low, and moderate certainty of the evidence, respectively (Table 2). Likewise, for the outcome of parasite prevalence a reduction was observed for the clusters that received chlorfenapyr-pyrethroid ITNs at 12-months (Fig 13), 18-months (Fig 14) and 24-months (Fig 15) post follow-up. These findings were associated with low, low, and moderate certainty in the evidence, respectively (Table 2).

For clusters that received pyriproxyfen-pyrethroid ITNs, the authors have reported that compared to clusters that received pyrethroid-PBO ITNs, there was an increase in malaria case incidence when collapsed across time points (Fig 23), at 1-year post intervention (Fig 24), and for 2-years post intervention (Fig 25). These findings were associated with moderate, moderate, and low certainty of the evidence (Table 4). This was also consistent for the outcome of parasite prevalence, where increases were observed for parasite prevalence in the clusters that received pyriproxyfen-pyrethroid ITNs at 12-months (Fig 26), 18-months (Fig 27), and 24-months (Fig 28) post follow-up compared to those that received pyrethroid-PBO ITNs. These findings were associated with low, moderate, and low certainty of the evidence, respectively (Table 4).

There are some limitations to these findings. Firstly, only three studies were identified that met the inclusion criteria of this review, however all these studies are recent (within the last 5 years) and have compared similar interventions, have implemented the interventions uniformly achieving high coverage, and have reported similar results. However, this does suggest limitations to the transferability of this data as the results have all come from high-transmissions settings with pyrethroid resistant *An. Gambiae s.l* and/or *An. Funestus s.l* vectors from Africa.

Secondly, major differences between these studies included the manufacturer of the ITNs compared, the main vectors of interest and their setting. However, upon conducting subgroup analyses (S4 in S1 File) and ICEMAN credibility assessments (S5 in S1 File) none of these factors were deemed to be effect modifiers. It is also important to note, that the analyses conducted were ill-suited to detect sub-group differences, due to so few studies being included, and each study having been conducted in different settings and with different dominant vector

species. As effect modification is viably plausible, we emphasise that while we did not detect effect modification from any of the investigated subgroups, uncertainty remains, and effect modification may still be present.

We argue that while these findings should be interpreted carefully within the context of a guideline panel, they should also be interpreted in relation to other endpoints assessed regarding the same comparison. For example, caution should be taken when interpreting the results presented in Fig 17 (pyriproxyfen-pyrethroid ITNs versus pyrethroid-only ITNs) in isolation from the results presented in Figs 16 and 18, as this result may suggest these DAI ITNs have superiority over pyriproxyfen-ITNs that may not exist when considering the entire body of evidence. Finally, the second review question that was initially asked was unable to be answered in this review with the data made available to the review team, as such, no conclusions have been made regarding this data. Future research is needed on these types of nets to investigate this concern.

All studies were cluster randomised controlled trials and therefore, the overall certainty in the body of evidence started as high. The impact of DAI ITNs has been evaluated in lower-level evidence, however this has not contributed to the evidence synthesised as part of this review. Publication bias was unable to be assessed during this review as only three studies were included. However, the comprehensive search strategy and contacting of authors directly, ameliorated some concerns of publication bias in this review.

## Conclusion

We have high certainty evidence that chlorfenapyr-pyrethroid ITNs are more effective than pyrethroid-only ITNs in reducing malaria case incidence. This benefit also extends to parasite prevalence for which we have moderate-high certainty evidence. However, only chlorfenapyr-pyrethroid ITNs demonstrated a reduction in these outcomes when compared to pyrethroid-PBO ITNs.

Despite most of this evidence being high-moderate certainty, only three studies were included in this review. These studies were conducted in high transmission settings, and additional studies conducted in other transmission settings would further strengthen the evidence base in favour of chlorfenapyr-pyrethroid DAI ITNs. Future trials should also explore these interventions for longer than 2-years post implementation of the intervention to provide more robust data as to their long-term effectiveness.

## Deviations from protocol

1. No need to adjust standard errors for failing to account for clustering as all studies had done so appropriately. Where raw data has been used, risk of bias implications have been taken into consideration.

2. The following subgroups were specified in the protocol but were not conducted for the stated reasons

   a. Level of transmission; (High: incidence of about 450 cases/1000 persons/year or Plasmodium falciparum (Pf) / Plasmodium vivax (Pv) prevalence of > = 35%; Moderate: incidence of 250–450 per 1000 persons per year and Pf/Pv prevalence of 10–35%; Low: incidence of 100–250 per 1000 persons per year and Pf/Pv prevalence of 1–10%; Very low: incidence of <100 per 1000 persons per year and Pf/Pv prevalence <1%.) (the level of transmission will be categorized according to the schema found in the *Framework for*

*malaria elimination)*; seasonality of transmission (Not conducted as all studies were from high transmission settings)

   b. Species of parasite (Not conducted as all studies explored P. falciparum)

   c. Coverage of intervention applied and level of net coverage per person or household (Not conducted as all studies had a high intervention coverage)

   d. Durability of net and insecticides used (No study had provided sufficient data regarding net durability. As the assessment of durability was not the focus of this review this sub-group has been omitted).

   e. Characteristics of insecticides used, e.g., target sites, modes of action, and duration required to produce such effect(s). (The subgrouping parameters of this pre-specified group were identical to the "insecticide class" subgroup that has been presented in the review. As such, these two subgroups have been combined and reported together in the review.

   f. Population demographics e.g., sex/age/SES/ethnicity etc. All included studies provided population demographics for the study cohorts. These demographics were similar enough to not warrant subgrouping).

   g. Human behaviour (e.g. sleeping behaviour) (Two studies had provided this information, however one study had not. After contacting the authors for this information, it was not received and the two remaining studies were not different enough to warrant subgrouping).

   h. Coverage of other background interventions. (All included studies confirmed that no other interventions were being carried out in the trial region during the study period. As such, no subgrouping necessary).

3. Sensitivity analyses was originally planned conducted to analyse the following (below). However, as all the included studies were at low risk of bias, and had appropriately controlled for clustering it was unnecessary.

   a. The impact of bias by excluding studies that are at a high risk of bias.

   b. Where we have inflated standard errors for trials where cluster designs have not been considered, we will analyse trials as if the individual was the unit of randomisation.

4. The following outcomes were not included in the GRADE evidence profiles due to a lack of available data from the included studies:

   a. Malaria infection incidence

   b. Incidence of severe disease

   c. All-cause mortality

   d. Malaria mortality

## Supporting information

**S1 Checklist. PRISMA 2020 checklist.**
(DOCX)

**S1 File.**
(DOCX)

## Acknowledgments

The authors would like to acknowledge and thank staff from the Global Malaria Programme, WHO, for their input and support. We would like to acknowledge Jan Kolaczinski, Head, Vector Control & Insecticide Resistance Unit, Global Malaria Program, WHO for Department of Internal Medicine, and Elie Akl, American University of Beirut, Lebanon for their guidance and feedback. We also acknowledge the feedback from the members of the Guideline Development Group for this project.

## Author Contributions

**Conceptualization:** Timothy Hugh Barker, Zachary Munn.

**Data curation:** Timothy Hugh Barker, Jennifer C. Stone, Sabira Hasanoff, Carrie Price, Alinune Kabaghe, Zachary Munn.

**Formal analysis:** Timothy Hugh Barker, Zachary Munn.

**Funding acquisition:** Timothy Hugh Barker, Zachary Munn.

**Investigation:** Timothy Hugh Barker.

**Methodology:** Timothy Hugh Barker, Jennifer C. Stone, Carrie Price, Alinune Kabaghe, Zachary Munn.

**Project administration:** Timothy Hugh Barker.

**Supervision:** Zachary Munn.

**Visualization:** Timothy Hugh Barker.

**Writing – original draft:** Timothy Hugh Barker, Jennifer C. Stone, Sabira Hasanoff, Zachary Munn.

**Writing – review & editing:** Timothy Hugh Barker, Jennifer C. Stone, Sabira Hasanoff, Alinune Kabaghe, Zachary Munn.

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
