## [Decision Letter · Decision Letter 0]

24 Apr 2023

PONE-D-23-09253Effectiveness of dual active ingredient insecticide-treated nets in preventing malaria: a systematic review and meta-analysisPLOS ONE

Dear Dr. Barker,

Thank you for submitting your manuscript to PLOS ONE. After careful consideration, we feel that it has merit but does not fully meet PLOS ONE’s publication criteria as it currently stands. Therefore, we invite you to submit a revised version of the manuscript that addresses the points raised during the review process. Please submit your revised manuscript by Jun 08 2023 11:59PM. If you will need more time than this to complete your revisions, please reply to this message or contact the journal office at plosone@plos.org. Please include the following items when submitting your revised manuscript:A rebuttal letter that responds to each point raised by the academic editor and reviewer(s). You should upload this letter as a separate file labeled 'Response to Reviewers'.A marked-up copy of your manuscript that highlights changes made to the original version. You should upload this as a separate file labeled 'Revised Manuscript with Track Changes'.An unmarked version of your revised paper without tracked changes. You should upload this as a separate file labeled 'Manuscript'.

We look forward to receiving your revised manuscript.

Kind regards,

James Colborn

Academic Editor

PLOS ONE

Journal Requirements:

Additional Editor Comments:

Both reviewers found the manuscript to be overall well-written, but both still did identify corrections that must be made before it can be published. More importantly Reviewer 1 raised serious concerns about the statistical methodology utilized, and more specifically the calculation and use of 95% confidence intervals. Given the implications of the results of this meta-analysis, these concerns must be addressed fully before the manuscript can be considered for publication.

Reviewers' comments:

Reviewer's Responses to Questions

**Comments to the Author**

1. Is the manuscript technically sound, and do the data support the conclusions?

Reviewer #1: No

Reviewer #2: Partly

2. Has the statistical analysis been performed appropriately and rigorously? 

Reviewer #1: No

Reviewer #2: Yes

3. Have the authors made all data underlying the findings in their manuscript fully available?

Reviewer #1: Yes

Reviewer #2: Yes

4. Is the manuscript presented in an intelligible fashion and written in standard English?

Reviewer #1: Yes

Reviewer #2: No

5. Review Comments to the Author

Reviewer #1: EFFECTIVENESS OF DUAL ACTIVE INGREDIENT INSECTICIDE-TREATED NETS IN PREVENTING MALARIA: ASYSTEMATIC REVIEW AND META-ANALYSIS

This meta-analysis and systematic review investigated two types of dual active-ingredient (DAI) insecticide-treated nets (ITN) for malaria prevention. These are the first ITNs that contain a second active ingredient in addition to the pyrethroid insecticide found in standard nets. The purpose of the second active ingredient is to kill or render sterile female mosquitoes that survive contact with pyrethroid insecticide due to resistance. If proven superior to the standard pyrethroid only ITN, these nets will fill a major gap in the arsenal of tools for the control of malaria in areas of pyrethroid resistant, which is widespread across Africa. If incorrectly deemed superior to the standard net, which is significantly cheaper than the new nets, there will be no gain resulting from the switching to the new net, but a considerable downside as the increased cost of the new net could be offset by smaller volumes being procured leading to lower overall coverage. A pre-print pre-peer review version of this systematic review formed the basis of evidence considered by a WHO panel for formulating guidance on whether these new nets should be deployed across sub-Saharan Africa. Accordingly, on 14 March 2023, WHO published “recommendations on two new types of insecticide-treated nets”, i.e. the nets that form the subject of this systematic review. No doubt the authors of this manuscript, and those of us tasked with peer-reviewing their work, will therefore be aware of the particular responsibility that is placed upon us.

MAJOR COMMENTS

1. In light of the above I regret to say that the analysis underpinning this review contains a major flaw that is likely to invalidate at least some of the conclusions they draw. All three trials that contributed to this review were cluster randomised, and the authors correctly state that analysis of such data needs to take account of their clustered nature in calculating standard errors (SEs) and hence 95% Confidence Intervals. The authors note that there was “No need to adjust standard errors for failing to account for clustering as all studies had done so appropriately”. Unfortunately they then do not use the appropriate 95%CI that were reported in the studies and which allow for appropriate inflation of SEs, but calculate their own, so-called ‘exact 95% CI’ from the crude numbers given in the tables of the papers without taking account of the clustered nature from which these numbers are aggregated. I checked a number of 95% CIs that are cited in the review and all were identical to what one would get from assuming a simple random sample of independent observations, rather than a set of observations that are correlated within clusters. It is unclear why the 95% CIs that are cited in the papers, and which the authors confirm as taking account of clustering, were replaced by the narrower CIs that ignore clustering. Where data are pooled across studies, it appears that 95% CI were computed equivalent to adding the corresponding cases and denominators from the constituent studies, and calculating SEs assuming simple binomial or Poisson distributions (depending on outcome type), with no between cluster variance component. As a consequence, all 95% CIs are too narrow and need to be recalculated.

In some cases the corrected results are likely not to lead to a different interpretation regarding the superiority of the new nets, in others it is highly likely that a different conclusion will need to be drawn. For example, for the Royal Guard pyriproxyfen net the two papers that contribute outcomes for malaria incidence (Accrombessi et al and Mosha et al) for the comparison with the standard pyrethroid net, the overall results cited in each of the two papers, are given as incidence rate ratios [95% Cis] of 0·86 [ 0·65–1·14]. p= 0·28 (Accrombessi) and 0·99 (0·66–1·50), p=0·9801 (Mosha), i.e. no evidence against the null hypothesis of no effect in each trial alone. This changes dramatically if one calculates exact 95% CIs that ignore the lack of independence in the data (Fig 25 of the manuscript), and the authors come up with a highly significant superior protection afforded by this net, with an IRR of 0.86 [0.78- 0.94]. It is likely that the corrected result would show little or no evidence of superiority of Royal Guard over the standard pyrethroid net once the analysis takes account of clustering. Therefore, erroneously this net has been recommended as providing better protection based on ‘strong evidence’.

The conclusion “the evidence also suggests that clusters that receive a pyriproxyfen-pyrethroid ITN (figure 25) will result in a reduction of malaria case incidence compared to clusters that receive a pyrethroid-only ITN.... This finding was associated with high certainty of the evidence” (page 43) therefore seems highly unlikely. This error can have important practical consequences, since international donors purchase nets based on WHO guidance. Therefore, in addition to correcting the meta analysis for this paper, it is incumbent on the authors to contact WHO to inform them that their recently promulgated recommendation has been based on a flawed analysis and may need to be withdrawn.

The errors are simple to remedy and I am sure the authors need no advice on this: if the correct 95% CIs cited in the published papers together with the raw numbers of cases and denominators were entered into the meta-analysis software, it will calculate the appropriate inflation that is applied to the standard errors for each study and hence apply this to the pooled analysis – I am certain the authors will know exactly what needs to be done (and probably wonder how this slip-up occurred). However, in its current form the paper cannot be published, and the discussion and conclusion will need to reflect a revised set of results, possibly contradicting some of the current recommendations.

As the authors are highly experienced in this type of work, this is probably just a glitch, but one with significant consequences for the findings.

2. Apart from this major issue with standard errors, the meta-analysis has been meticulously carried out.

3. The meta-analysis also assessed variation of effect in subgroups (effect modification) to see whether efficacy is affected by factors such as mosquito vector species, urbanicity etc. No such effects were evident, but the discussion should acknowledge that this analysis is ill-suited for such an investigation. These factors were specific to each study site with little or no variation within each study area. For example, effect modification by vector species cannot be assessed given that each vector is only present in either the one or the other site. Hence any potential effect modification would be entirely subsumed by differences between studies.

Since effect modification is biologically plausible, there should be acknowledgement that the analysis had low power for detecting this if it existed.

4. Combining the results of the three trials, two that used Royal Guard pyriproxyfen nets (cited above) and one that investigated the Sumitomo Olyset Duo pyriproxyfen net (Tiono et al), is of little value and possibly misleading given that the latter net has been withdrawn from the market some time ago by the manufacturers. Any conclusions based on a corrected analysis should make clear that the Royal Guard net (the only pyriproxyfen net currently available) failed to show superiority to standard nets in two independent trials.

It should also be mentioned that Olyset Duo is not a WHO pre-qualified net, see

https://extranet.who.int/pqweb/vector-control-products/prequalified-product-list?field_product_type_tid=100&field_pqt_vc_ref_number_value=&title=&field_applicant_tid=&field_active_ingredient_synergis_tid=

MINOR COMMENTS

1. Eligibility criteria should include proven pyrethroid resistance in malaria vectors in the study area (summary and p 26).

2. Introduction, first paragraph: cite figures on global malaria burden from most recent World Malaria Report, rather than outdated ones.

3. Introduction, first paragraph: malaria case incidence should be expressed per 1000 persons at risk.

4. Also Introduction, first paragraph: It is incorrect to refer to the use of ITNs as a treatment strategy. It is a strategy for prevention (or control, not treatment).

5. Introduction, second paragraph, second sentence: this statement is dated: all nets are now factory treated long lasting insecticidal nets and those distributed through government or NGO agencies do not require re-treatments. Self treatment kits are generally no longer used.

6. Page 25: There seems to be a muddle as to what are the same and what are different classes of insecticide. Some rephrasing is needed here.

7. Page 26, objectives: As stated, these objectives go beyond what was actually addressed in this study, namely improved efficacy against malaria infection. The question whether something should actually be used rather than something else, is broader, including cost, coverage implications, acceptability, durability, etc. The objectives should be narrowed to what was actually investigated in this study, namely efficacy as a method of preventing malaria infection, compared to the current standard of care.

8. Objectives (p 26): The term “long lasting insecticidal nets” should be used in both objectives – not just objective two.

9. Page 27, line 2: State: population at risk for specified duration, or cases over person time.

10. Page 27, definition of mortality of adult female Anopheles should be widened to cover mortality caused by slow acting insecticides such as chlorfenapyr, since mortality due to exposure to these may take up to 72 hours.

11. Page 30, unit of analysis: “ the inflation of standard errors was not required as originally planned as per the protocol”. This does not mean it can be ignored, see point 1 of major comments.

12. Page 31, Role of funding source. “The funder of the study had a role in the development of the protocol, the wording and development of the review questions, the interpretation of the final results and the development of this manuscript.” This seems rather unusual – Note to EDITOR: is it in line with the policy of the journal?

13. Page 32, Interventions: Nets are not 'coated' with insecticide. “Impregnated” would be a better description, since the chemical is intended to be gradually released to the surface of the fibre over the lifetime of the net.

14. Page 32, description of Accrombessi study: “ Both interventions were compared against each other”. This is incorrect: the interventions were only compared to control, not to each other. Not comparing to each other is a deliberate part of the design of the study to reduce the frequency of statistical testing. This also applies to the Mosha et al study which adjusted p-values to take account of multiple testing. Inventing additional tests post hoc invalidates this approach. The Olyset-PBO net was only compared to control, not to the other two interventions as stated in the text. The authors should correct this description. This does not prevent the comparisons being made in this secondary analysis of the data.

In both studies (Accrombessi et al and Mosha et al) the comparator was the Interceptor net – in the text it appears that this is only the case for Mosha et al.

15. Page 32, description of different nets: It would be better to give specs about chemical content in the same way for all nets, e.g. g/m2.

16. Page 35, Bias arising from randomisation process. Suggest rewording this to avoid restricted randomisation appearing to be muddled with allocation concealment -- the two have different objectives and are separate processes. Not doing the one does not affect the other and vice versa.

Reviewer #2: The paper presents a Cochrane review and meta-analysis of three independent cluster randomized trials investigating the incremental impact of dual-AI ITNs relative to standard, pyrethroid-only ITNs.

Some general questions and comments include:

Summary:

• The number of incident cases averted per 1000 person-years at risk is a great way to frame the results, but an IRR would also be quite helpful for additional context. What were the actual incidence rates in the intervention clusters compared to the standard of care clusters? How big of a relative (i.e., percent) reduction is 190 and 145 fewer per 1000 person-years at risk?

• The interpretation could be better contextualized, as even though Dual AI ITNs do demonstrate consistent reductions in malaria case incidence and other outcomes across multiple comparisons, this statement is not explicit about what those comparison are. I think it is important to be specific and acknowledge that Dual AI ITNs provided better control of malaria than standard, pyrethroid-only ITNs in a variety of settings with pyrethroid resistant vectors.

Summary of Finings Table:

• The descriptions of the Participants Follow-up (the first column) could be clearer. Specifically, how many studies are included in each assessment (for example, in the first row is it 3 total studies, 2 RCTs with 2000-person years plus another study with 61,183 participants?).

• For the Anticipated absolute effects risk difference calculations (the last column), can you clarify if this is presented as a mean estimate number of cases averted per 1,000 with a 95% credibility interval? Also, maybe consider providing a brief explanation of the calculation and the source of the IRR used for this in the footnotes? It’s not immediately clear why sometimes the risk difference is straightforward (the difference in the event rates across study arms) and sometimes it is not (e.g., the first row of the chlorfenapyr-pyr ITNs compared to pyr-only ITNs table).

•Would be nice to standardize the number of significant figures presented throughout the table as well…not sure that carrying out the case incidence rates to the hundredths of a case per 1000 person-year is necessary.

Introduction:

• This section needs to be proofread and cleaned up, and could be more focused than currently presented. Some examples of things to consider include:

o Is P. vivax usually considered one of the more virulent Plasmodium spp. parasites to infect humans?

o ITNs (and vector control in general) are not treatment strategies, but prevention strategies.

o I think the last sentence of the introductory paragraph is incomplete.

o I know that the classification scheme for ITNs has recently changed, but I do not think that it is correct to describe pyrethroid-only nets as requiring treatment with a treatment kit of periodic re-treatment…this might be confusing newer and older WHO terminologies.

o Based in part on the results reviewed in this meta-analysis, the public health value of DAI nets has been determined…I think it is appropriate to change the verb tense of this sentence and provide some more context. Perhaps “The public health value and incremental cost-effectiveness of Dual AI ITNs had not been established until recently.”

o The final sentences of the last paragraph on the first page of the intro does not fit well with the first few sentences – there is no good transition from the discussion of how pyriproxyfen works to the problem of pyrethroid resistance.

o I’m not sure the review questions are appropriately framed, as they do not include a discussion of pyrethroid resistance in the local vectors “In areas with ongoing malaria transmission,” which were key considerations during each of the randomized trials reviewed. The question is a bit more complex than framed, and might be better if described like: “In areas with ongoing malaria transmission and pyrethroid resistance vectors, should nets treated with a pyrethroid and non-pyrethroid insecticide…”

o Also, there are some interesting policy implications in the language used: “should” dual AI nets be used “versus” standard nets. This would require a robust cost-effectiveness analysis, and a discussion of current WHO GMP recommendations on universal coverage targets. The questions are more typically framed like “Do Dual AI ITNs provide additional protection from malaria transmission, compared to standard pyrethroid-only ITNs, in areas of active malaria transmission and pyrethroid resistant vectors…”

Results:

• The Accrombessi paper from Benin has since been published and the text could be updated to reflect this.

• It would be helpful if table 6 included a column with a basic description of the pyrethroid resistance status of the primary vector in each study location.

Discussion:

• I think there may be a mix-up in the second paragraph, as Table 1 describes a high certainty of evidence that chlorfenapyr-pyrethroid ITNs will results in a reduction in malaria case incidence compared to pyrethroid-only ITNs, but the narrative describes this as only Moderate certainty.

6. PLOS authors have the option to publish the peer review history of their article (what does this mean?). If published, this will include your full peer review and any attached files.

Reviewer #1: No

Reviewer #2: No

---

## [Author Response · Author response to Decision Letter 0]

9 May 2023

Please see the attached document "Response to Reviewers" for a detailed response to every comment left by the peer-reviewers and the editor.

---

## [Decision Letter · Decision Letter 1]

25 Jun 2023

PONE-D-23-09253R1Effectiveness of dual active ingredient insecticide-treated nets in preventing malaria: a systematic review and meta-analysisPLOS ONE

Dear Dr. Barker,

Thank you for submitting your manuscript to PLOS ONE. After careful consideration, we feel that it has merit but does not fully meet PLOS ONE’s publication criteria as it currently stands. Therefore, we invite you to submit a revised version of the manuscript that addresses the points raised during the review process. The reviewers thank the authors for their comprehensive responses to their comments, though they still have some concerns. Most additional comments the reviewers have are minor, though those raised by reviewer 2 in particular must be addressed prior to publication.

We look forward to receiving your revised manuscript.

Kind regards,

James Colborn

Academic Editor

PLOS ONE

Reviewers' comments:

Reviewer's Responses to Questions

**Comments to the Author**

1. If the authors have adequately addressed your comments raised in a previous round of review and you feel that this manuscript is now acceptable for publication, you may indicate that here to bypass the “Comments to the Author” section, enter your conflict of interest statement in the “Confidential to Editor” section, and submit your "Accept" recommendation.

Reviewer #1: (No Response)

Reviewer #2: (No Response)

2. Is the manuscript technically sound, and do the data support the conclusions?

Reviewer #1: Yes

Reviewer #2: Partly

3. Has the statistical analysis been performed appropriately and rigorously? 

Reviewer #1: Yes

Reviewer #2: Yes

4. Have the authors made all data underlying the findings in their manuscript fully available?

Reviewer #1: Yes

Reviewer #2: Yes

5. Is the manuscript presented in an intelligible fashion and written in standard English?

Reviewer #1: Yes

Reviewer #2: Yes

6. Review Comments to the Author

Reviewer #1: Thank you for responding to my previous comments so thoughtfully and for having addressed them where appropriate.

I have only two remaining points that should be dealt with before publication:

1. Having taken care of the inflation of standard errors and 95% CIs to account for the within-cluster correlation of responses after my earlier comments relating to this (and now reflecting this in the revised results), I don't see this mentioned anywhere in the methods section of the manuscript. Please add a sentence stating that calculation of 95%CI took account of the clustered nature of the data.

Apologies if it is there and I have missed it.

2. Overall there are a very large number of endpoints in this paper and it is therefore possible that one or two results may be due to chance, despite the fact that these data derive from Randomised trials. This is particularly a strong possibility were one endpoint result is contrary to the pattern of results for a particular intervention. For example, the pyrethroid-insect growth regulator (pyriproxyfen) dual active ingredient vs pyrethroid only net comparison shows 95% CIs overlapping 1 (i.e. non significant) for all but one endpoint, namely malaria case incidence after 12 months. It is highly implausible that this net would be superior to pyrethroid only nets in this one particular way but otherwise no better than the standard net. A more plausible interpretation is that this result is a chance finding and there is little evidence that pyriproxyfen DAI nets are superior to standard nets (the null hypothesis). There should thus be some caution expressed that multiple testing will lead to chance findings and it is therefore important to interpret results not in isolation but in relation to other endpoints for the same product.

Apart from this everything looks good to me. Congratulations with an important piece of work!

Reviewer #2: The revisions have addressed many of the comments and concerns raised by the reviewers and editors, particularly the re-calculation of the confidence intervals, the reframing the prevalence results as odds ratios, and the re-interpretation of many of the conclusions in light of the updated analysis.

Most of the revised text is also easier to understand and improves the readability of the paper.

The methods, analysis, and assessment of heterogeneity and risk of bias all seem robust and well carried out. Unfortunately, I am still concerned that the review questions are not framed appropriately and the conclusions are too broad as presented, and can not recommend publication as written.

Appreciating that the GMP provided the research questions to the review team, they are still poorly framed and not addressed by the studies currently available.

For question 1, the available data only address transmission settings with pyrethroid resistant An. gambiae s.l. and/or An. funestus s.l. in Africa. The question, and the conclusions of the paper, need to be revised to reflect this in order to recommend publication. Also, I believe that WHO PQ prefer the original term used “nets treated with,” or the term “insecticide treated nets (ITNs) utilizing a pyrethroid…”compared to the the revised phrase “long last insecticidal nets treated with….”

For question 2, the question as written is simply not answerable with the data available. It is comforting to see that no recommendation or conclusion relative to this question is presented in the paper, and there is some acknowledgment of the nuance required when interpreting the results in the context of malaria control programming. However, the second research question should be removed or, alternatively, acknowledged as not yet answerable while highlighting the knowledge gaps that need to be addressed in order to do so (e.g., cost effectiveness – in the face of a volatile ITN market with shifting net prices and new tools on the horizon, and a need to revisit current WHO GMP policy recommendations with regard to universal coverage targets).

7. PLOS authors have the option to publish the peer review history of their article (what does this mean?). If published, this will include your full peer review and any attached files.

Reviewer #1: **Yes: **Professor Immo Kleinschmidt

Reviewer #2: No

---

## [Author Response · Author response to Decision Letter 1]

30 Jun 2023

Reviewer #1 

Thank you for responding to my previous comments so thoughtfully and for having addressed them where appropriate. I have only two remaining points that should be dealt with before publication:

1. Having taken care of the inflation of standard errors and 95% CIs to account for the within-cluster correlation of responses after my earlier comments relating to this (and now reflecting this in the revised results), I don't see this mentioned anywhere in the methods section of the manuscript. Please add a sentence stating that calculation of 95%CI took account of the clustered nature of the data.

Apologies if it is there and I have missed it.

We thank the reviewer for identifying this oversight in our methods. We have added the following sentence to lines 333-334 of the “data synthesis and meta-analysis sections of the revised manuscript. “Calculation of 95% CIs took account of the clustered nature of the data where appropriate.”

2. Overall there are a very large number of endpoints in this paper and it is therefore possible that one or two results may be due to chance, despite the fact that these data derive from Randomised trials. This is particularly a strong possibility were one endpoint result is contrary to the pattern of results for a particular intervention. For example, the pyrethroid-insect growth regulator (pyriproxyfen) dual active ingredient vs pyrethroid only net comparison shows 95% CIs overlapping 1 (i.e. non significant) for all but one endpoint, namely malaria case incidence after 12 months. It is highly implausible that this net would be superior to pyrethroid only nets in this one particular way but otherwise no better than the standard net. A more plausible interpretation is that this result is a chance finding and there is little evidence that pyriproxyfen DAI nets are superior to standard nets (the null hypothesis). There should thus be some caution expressed that multiple testing will lead to chance findings and it is therefore important to interpret results not in isolation but in relation to other endpoints for the same product.

We agree with the reviewer and have added a few sentences regarding this point to the limitations section of our discussion. The following sentences have been added to lines 955-960 of the revised manuscript. “Finally, we argue that while these findings should be interpreted carefully within the context of a guideline panel, they should also be interpreted in relation to other endpoints assessed regarding the same comparison. For example, caution should be taken when interpreting the results presented in figure 17 (pyriproxyfen-pyrethroid ITNs versus pyrethroid-only ITNs – 12 months) in isolation from the results presented in figure 16 and 18, as this result may suggest these DAI ITNs have superiority over pyriproxyfen-ITNs that may not exist when considering the entire body of evidence.”

Reviewer #2

The revisions have addressed many of the comments and concerns raised by the reviewers and editors, particularly the re-calculation of the confidence intervals, the reframing the prevalence results as odds ratios, and the re-interpretation of many of the conclusions in light of the updated analysis.

Most of the revised text is also easier to understand and improves the readability of the paper.

The methods, analysis, and assessment of heterogeneity and risk of bias all seem robust and well carried out. Unfortunately, I am still concerned that the review questions are not framed appropriately and the conclusions are too broad as presented, and can not recommend publication as written.

Appreciating that the GMP provided the research questions to the review team, they are still poorly framed and not addressed by the studies currently available.

For question 1, the available data only address transmission settings with pyrethroid resistant An. gambiae s.l. and/or An. funestus s.l. in Africa. The question, and the conclusions of the paper, need to be revised to reflect this in order to recommend publication. Also, I believe that WHO PQ prefer the original term used “nets treated with,” or the term “insecticide treated nets (ITNs) utilizing a pyrethroid…”compared to the the revised phrase “long last insecticidal nets treated with….”

For question 2, the question as written is simply not answerable with the data available. It is comforting to see that no recommendation or conclusion relative to this question is presented in the paper, and there is some acknowledgment of the nuance required when interpreting the results in the context of malaria control programming. However, the second research question should be removed or, alternatively, acknowledged as not yet answerable while highlighting the knowledge gaps that need to be addressed in order to do so (e.g., cost effectiveness – in the face of a volatile ITN market with shifting net prices and new tools on the horizon, and a need to revisit current WHO GMP policy recommendations with regard to universal coverage targets).

We thank the reviewer for their time reviewing this manuscript and their suggestions to improve the quality of the review. While we have actioned some changes (outlined below), we will not be removing or adjusting the questions as presented in the review. These questions were provided to the review team from the WHO Global Malaria Programme and to change them or remove them at this stage of the review process would not be appropriate. We hope the reviewer can appreciate and accept this. We will, however, change the wording of the questions from “long lasting insecticidal nets…” to “insecticide-treated nets” as this does not change the intent of the original question posed. We do agree with the review that more explicit mention can be made to the limitations of the review questions and how they relate to the conclusions of the paper. Firstly, we have made changes to the sentence from lines 945-947 that clarify that the data comes from high-transmission settings with pyrethroid resistant An. Gambiae s.l and/or An. Funestus s.l in Africa and therefore generalisability of these results may be hampered. “However, this does suggest limitations to the transferability of this data as the results have all come from high-transmissions settings with pyrethroid resistant An. Gambiae s.l and/or An. Funestus s.l vectors from Africa.” 

We have also added an additional sentence to the limitation’s sections of the discussion regarding the second review question. This change has been added from lines 960-962. “Finally, the second review question that was initially asked was unable to be answered in this review with the data made available to the review team, as such, no conclusions have been made regarding this data.”

---

## [Decision Letter · Decision Letter 2]

19 Jul 2023

Effectiveness of dual active ingredient insecticide-treated nets in preventing malaria: a systematic review and meta-analysis

PONE-D-23-09253R2

Dear Dr. Barker,

We’re pleased to inform you that your manuscript has been judged scientifically suitable for publication and will be formally accepted for publication once it meets all outstanding technical requirements.

Kind regards,

James Colborn

Academic Editor

PLOS ONE

Additional Editor Comments (optional):

Reviewers' comments:

Reviewer's Responses to Questions

**Comments to the Author**

1. If the authors have adequately addressed your comments raised in a previous round of review and you feel that this manuscript is now acceptable for publication, you may indicate that here to bypass the “Comments to the Author” section, enter your conflict of interest statement in the “Confidential to Editor” section, and submit your "Accept" recommendation.

Reviewer #1: All comments have been addressed

Reviewer #2: All comments have been addressed

2. Is the manuscript technically sound, and do the data support the conclusions?

Reviewer #1: Yes

Reviewer #2: Yes

3. Has the statistical analysis been performed appropriately and rigorously? 

Reviewer #1: Yes

Reviewer #2: Yes

4. Have the authors made all data underlying the findings in their manuscript fully available?

Reviewer #1: Yes

Reviewer #2: Yes

5. Is the manuscript presented in an intelligible fashion and written in standard English?

Reviewer #1: Yes

Reviewer #2: Yes

6. Review Comments to the Author

Reviewer #1: (No Response)

Reviewer #2: Thank you for your careful consideration of the suggestions, and for working to address them to the extent possible. The most recent revisions are helpful at defining and acknowledging some of the limitations of the data.

7. PLOS authors have the option to publish the peer review history of their article (what does this mean?). If published, this will include your full peer review and any attached files.

Reviewer #1: **Yes: **Immo Kleinschmidt

Reviewer #2: No

---

## [Editor Report · Acceptance letter]

24 Jul 2023

PONE-D-23-09253R2 

Effectiveness of dual active ingredient insecticide-treated nets in preventing malaria: a systematic review and meta-analysis 

Dear Dr. Barker:

I'm pleased to inform you that your manuscript has been deemed suitable for publication in PLOS ONE. Congratulations! Your manuscript is now with our production department. 

Kind regards, 

on behalf of

Dr. James Colborn 

Academic Editor

PLOS ONE